environmental science/environmental engineering/geochemistry

solidification and stabilization, field rainfall test, semi-dynamic leaching test, diffusion coefficient

**Authors for correspondence:**
Daofang Zhang
e-mail: zhangdf-usst@163.com
Feipeng Li
e-mail: leefp@126.com
Huancong Shi
e-mail: hcshi@usst.edu.cn

# The rainfall effect onto solidification and stabilization of heavy metal-polluted sediments

Yan Sun[1], Daofang Zhang[1], Feipeng Li[1], Hong Tao[1], Moting Li[1], Lingchen Mao[1], Zhujun Gu[1], Ziyang Ling[2] and Huancong Shi[1]

[1]Institute of Environment and Architecture, University of Shanghai for Science and Technology, Shanghai 200093, People's Republic of China
[2]School of Chemistry, The University of Manchester, Oxford Road, Manchester M13 9PL, UK

(iD) HS, 0000-0003-4333-4118

Rainfall makes impacts on the process of solidification/stabilization (S/S) and the long-term safety of solidified matrix. In this study, the effect of rainfall on solidification/stabilization process was investigated by the rainfall test. The unconfined compressive strength (UCS) and toxicity characteristic leaching procedure (TCLP) were adopted to characterize the properties of S/S sediments before and after the rainfall test. The samples cured for 28 days were selected for semi-dynamic leaching tests with a simulated acidic leachant prepared at pH of 2.0, 4.0 and 7.0. The effectiveness of S/S treatment was evaluated by diffusion coefficient ($D_e$) and leachability index (LX). The results indicated that UCS decreased at maximum deterioration rate of 34.23% after 7 days of curing, along with the minimum rate of 7.98% after 28 days by rainfall, with greater than 14 days referred. The rainfall had little effect on the leaching characteristics of heavy metals during the curing process. However, the simulated acid rain made significant impacts on the leaching behaviours of the heavy metals in the S/S materials. All the values of cumulative fraction of leached heavy metals were less than 2.0%, exhibition of good stabilization of cement. Furthermore, the calculated diffusion coefficient ($D_e$) for Cu was $1.28 \times 10^1 \, cm^2 \, s^{-1}$, indicating its low mobility of heavy metal ions in S/S sediments. Furthermore, the calculated diffusion coefficients ($D_i$) for Cd, Cu and Pb were $7.44 \times 10^{-11}$, $8.18 \times 10^{-12}$ and $7.85 \times 10^{-12} \, cm^2 \, s^{-1}$, respectively, indicating their relatively low mobility of heavy metal in S/S sediments.

# 1. Introduction

The contamination of river sediments, especially heavy metals, poses a serious threat to the environment in China [1,2]. It has become an important topic in the field of environmental protection to mitigate the secondary pollution of the sediment and adoption of the dredged sediment as a resource. The 22nd World Dredging Conference (Shanghai in 2019) reached consensus on the sustainable development of dredging, and proposed that dredging and its utilization of dredging mud should follow the principles of 'construction with nature' and 'optimization of ecological value'. The utilization of dredging soil resources can be recycled and re-used in the dredging project in an economical way and reduce the consumption of new materials. With the development of technology, the concept of sediment as a resource has drawn more and more attention. The stabilization/solidification (S/S) of heavy metal contamination using cement-based binders is one of the most effective methods to reduce the mobility of heavy metals in sediments [3]. After S/S treatment, the heavy metals are stored in the sediments matrix as metal-hydrated phases, calcium–metal compounds and metal hydroxides [4]. The heavy metals in sediments are converted into a less soluble, mobile or toxic form through physical and chemical treatment, such as encapsulation, precipitation, adsorption and complexation [5,6].

When exposed to the complex environmental conditions, the stability of the solidified sediments will be weakened by various factors such as rainfall, carbonation and freeze–thaw cycling [7]. The production of carbonation has a positive impact on the mechanical strength of the solidified sediments if using the cement-based S/S [8,9], while acid rain has a particularly adverse impact on the long-term safety of solidified heavy metal contaminated sediments [10]. Acid rain is a strong corrosive medium, which contains not only $H^+$ but also other species of $NH_4^+$, $Mg^{2+}$ and $SO_4^{2-}$ [11]. Okochi et al. [12] discovered the anionic and cationic corrosive media such as $H^+$, $NH_4^+$, $SO_4^{2-}$, $NO_3^-$ and $Mg^{2+}$ in acid rain-induced erosion damage to cement-based materials such as mortar and concrete. If cement-stabilized sediments are exposed to the acid rain, the hydrogen ions ($H^+$) react with the cement hydration products spontaneously, such as portlandite ($Ca(OH)_2$) and calcium silicate hydrate (C–S–H) in the sediments matrix [13]. This corrosion has a negative effect on the solidified sediments structure [14]. Therefore, rainfall experiments were usually conducted in the field to simulate the effects of complex environment on the leaching and diffusive properties of heavy metals.

Before the S/S-treated sediment is adopted in field of landfill, it is necessary to investigate the effect of different rainfall conditions on its leaching behaviours. The semi-dynamic leaching test is used to simulate the migration and leaching performance of heavy metals in the solidified contaminated sediments. The experimental results can determine whether the leaching of heavy metals is under diffusion control. The observation diffusion coefficient of heavy metals can be calculated. These tests can evaluate the long-term safety of solidified contaminated sediments.

For semi-dynamic studies, Wang et al. [15] studied leaching behaviours of Pb under different pH conditions for 90 days. Moon & Dermatas [16] investigated the leaching characteristics of various metals from quick lime/fly ash stabilized soils with semi-dynamic leaching tests with 0.0014 N acetic acid as leachant (pH = 3.25). Xue et al. [17] conducted a 2-year semi-dynamic leaching test to investigate the leaching behaviour of lead in S/S waste. These studies suggest that diffusion is mostly 'controlling leaching mechanism' of heavy metals contained in S/S materials.

The objective of this study was to investigate the effects of rainfall onto mechanical properties and leaching characteristics of cement-based solidified sediments. A series of rainfall tests were conducted on treated sediment samples to study the effects of rainfall on the leachability and mechanical properties. Simultaneously, the semi-dynamic leaching tests were performed on treated sediment samples with simulated acid rain (SAR) as the extraction leachant with initial pH values of 2.0, 4.0 and 7.0. The leaching data for selected heavy metals (Cu, Cd and Pb) were elaborated with a diffusional leaching model based on the penetration theory, which fitted the release mechanisms of these heavy metals and predicted the long-term leaching behaviours of cement-based solidified sediments.

# 2. Materials and experimental procedures

The experiments consist of three parts: (i) the preparation of specimens, including pretreatment of sediments and solidified samples; (ii) field rainfall tests and semi-dynamic leaching tests for detecting the behaviour of heavy metals migration under acid rainfall conditions; and (iii) evaluation of long-term safety assessment under rainfall conditions.

**Table 1.** The physical–mechanical properties of tested sediments.

| moisture content (%) | pH | particle size distribution (μm) | organic content (%) | liquid limit (%) | plastic limit (%) | plasticity index |
|---|---|---|---|---|---|---|
| 41.34 | 7.78 | 5–80 | 2.2 | 33.9 | 22.6 | 11.3 |

**Table 2.** The total amount of heavy metals of tested sediments and cement.

| test indicators | Cd | Cr | Cu | Ni | Pb | Zn |
|---|---|---|---|---|---|---|
| original sediment (mg kg$^{-1}$) | 0.429 | 53.0 | 29.2 | 26.1 | 32.0 | 110 |
| pretreated sediment (mg kg$^{-1}$) | 0.974 | 66.8 | 60.4 | 50.9 | 156 | 160 |
| cement (mg kg$^{-1}$) | 0.135 | 19.6 | 24.4 | 12.1 | 26.5 | 30.2 |

## 2.1. Materials

The sediments used in this study were taken from Chongming District, Shanghai (E 121°17′38.6658″, N 31° 46′26.6154″). A columnar sediment sampler was used to collect samples. After removing the lower and upper parts, the sediment in the middle of the sampling column (about 20 cm) was stored, and brought back to the laboratory for analysis. The properties of tested sediments are listed in table 1. The moisture content was measured by Halogen moisture meter HB43-S. The pH is measured per ASTM D492 using a pH meter HAD-421 [18]. The Atterberg limits are measured per ASTM D4318 [19]. Sediment texture can influence the pore structure formation and strength development [20]. In this study, the particle size distribution was measured by the laser particle size analyser (BT-9300z, Baite). The tested sediment consists of 3.7% clay particles, 70.9% silt particles, 25.4% sand particles and then classified as silty mud. The sediment is defined as a silt for this study. For measuring the organic matter content in sediment, 20.0 g sediment samples were dried at 60°C to equilibrium and then calcined in a muffle furnace at 500°C for 24 h. By calculating the weight loss before and after the process, the organic matter content of sediment was not high (2.2%), so the effect of organic matter was not considered in this research.

The cement was PO42.5 Portland cement (PC), purchased from Conch Cement (Shanghai) Co. Ltd. The purity of the commercially available chemical reagents was analytical grade of 99%, and they were $CH_3COOH$, $HNO_3$, $HClO_4$, HF and magnesium sulfate anhydrous.

To prepare contaminated sediment specimens, predetermined volume of $Pb(NO_3)_2$, $Cu(NO_3)_2 \cdot 3H_2O$ and $Cd(NO_3)_2 \cdot 4H_2O$ solution was added to the air-dried sediments until its water content reached 45%. The concentrations of the heavy metals added were based on the soil environmental quality risk control standard for soil contamination of agricultural land (China, GB 15618—2018). The reason for the selection of nitrate is that it is chemically inert compared with chloride, sulfate and acetate ions when reacting with the metal ion in cement hydration [21]. The concentrations of these nitrates were relatively low for this test, and the negative effect was not strong for delay of the setting and the hardening of the cement and treated sediments. Cd, Cu and Pb were selected as the representative research objects in heavy metal pollution. Other metals, such as Cr, Ni and Zn, were also representative with the content provided in table 2. The concentrations of heavy metals in sediments and cement are presented in table 2.

The sediments and solution were thoroughly mixed for about 20 min with an electric mixer to create the homogeneous slurry. After maintenance in a closed container for 15 days at a temperature of 20 ± 2°C, the sediments were evaporated at 60°C under water bath, and the evaporation was not stopped until the water content reached 37–38%.

For analysing the initial concentration of selected heavy metals, tested sediment samples were homogenized and dried at 105°C and then ground to pass a 100 mesh sieve. Then 0.2 g sediment sample was added into a PTFE digestion tank placed on the hot plate. HCl, $HNO_3$, HF and $HClO_4$ were added into the tank in order. After complete digestion, the residual solution was carefully collected and diluted, and then passed through a PTFE filter with a size of 0.45 μm, then acidized with concentrated $HNO_3$ for inductively coupled plasma mass spectrometry (ICP-MS) (Perkin Elmer Nexion 300x) analysis.

The prepared contaminated sediments and cement were placed in a plastic bottle and thoroughly mixed at a speed of 120 r.p.m. for 3 min firstly, and then mixed at a speed of 60 r.p.m. for 1 min to achieve homogeneity. The quality ratio was 1 : 4 of cement to the contaminated sediments. The mixture was then

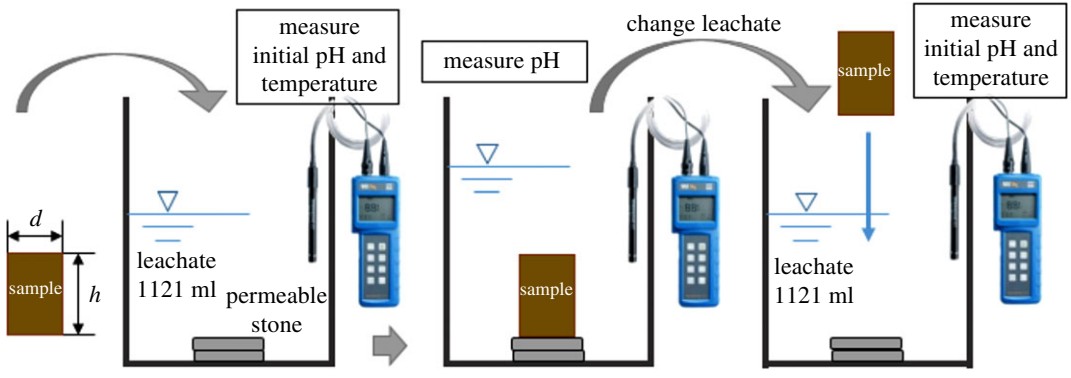

**Figure 1.** Schematic diagram for semi-dynamic leaching tests.

**Table 3.** The amount of rainfall and pH.

| the age of curing | 3 days | 7 days | 14 days | 28 days |
|---|---|---|---|---|
| the amount of rainfall (mm day$^{-1}$) | 55.9 | 46.6 | 65.5 | 46.8 |
| rainfall level | heavy rain | heavy rain | heavy rain to rainstorm | heavy rain |
| pH | 5.4 | 6.5 | 5.8 | 4.5 |

poured into a cylindrical mould with the diameter of 50 mm and height of 50 mm. After 24 h preservation, the sediment samples were carefully extruded from the mould with a hydraulic jack and subjected to standard curing condition (95% relative humidity and 22°C) for 3, 7, 14 and 28 days.

## 2.2. Experimental procedures

### 2.2.1. Field rainfall test

The field rainfall test was performed onto the standard cured samples in a beaker for different periods under natural rainfall conditions. From the beginning to the end of rainfall, the average period of the tests was 8 h. The amount of rainfall was between 25 and 50 mm (heavy rain) or 50 and 100 mm (rainstorm). Table 3 shows the actual pH and the amount of rainfall during the rainfall test. The detailed experimental procedures are provided in the electronic supplementary material.

Unconfined compressive strength (UCS) is an important index to measure the curing effect and evaluate the engineering properties of the cured body. Three groups of parallel experiments were conducted after 3, 7, 14 and 28 days of curing. The UCS tests as per ASTM D4219 [22] with a fixed strain rate of 1% min$^{-1}$ by means of a DYE-300S Model Compression and Fracture Resistance Integrative Machine (Huaxi Building Materials Experimental Instrument Co. Ltd). As the test started, the sediments sample was uniformly and continuously pressurized at the speed of 0.03–0.15 kN s$^{-1}$ until the break of the sample, and the break load pressure was recorded to calculate the UCS.

A certain amount of the fresh stabilized sediments was carefully sampled from the broken UCS specimen and then subjected to toxicity characteristic leaching procedure (TCLP) tests. The leachability of heavy metals is evaluated using the TCLP USEPA Method 1311 [23]. The initial pH was $4.93 \pm 0.05$ of the TCLP extraction liquid (5.7 ml of $CH_3COOH$ and 64.3 ml of $1 \, mol \, l^{-1}$ NaOH). The specimens were ground and sieved through a 2.0 mm screen. Then 10 g powder was stored in 200 ml of extraction liquid at a liquid/solid ratio of 20:1, and blended for 18 h on a rotary shaker at about 30 r.p.m. The leachate was filtered through a 0.45 µm membrane filter and stored in the refrigerator for determining concentrations of Cu, Pb and Cd with ICP-MS. Three parallel experiments were completed to ensure the reproducibility of the data, and the average value was presented as results.

### 2.2.2. Semi-dynamic leaching test

The semi-dynamic leaching test was performed as in figure 1, in order to investigate leaching behaviours and effectiveness of S/S-treated heavy metals contaminated sediments under different conditions.

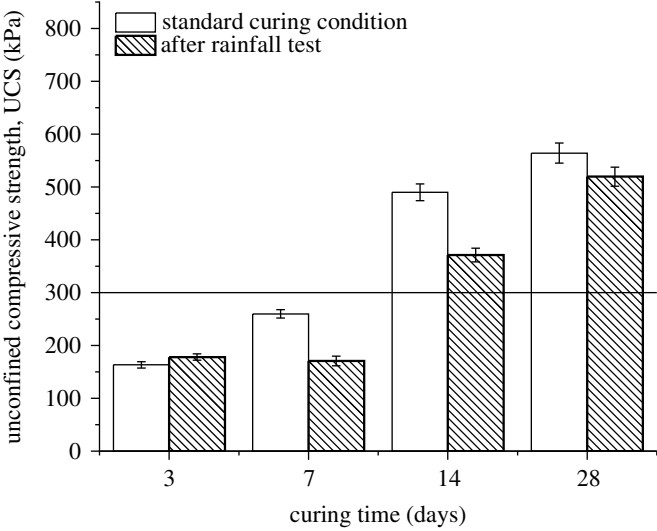

**Figure 2.** UCS comparison of standard curing condition and rainfall conditions, 300 kPa was selected as benchmark according to Resource Conservation and Recovery Act (RCRA) of the USA.

**Table 4.** The specific experimental conditions and test times for different tests.

| test type | | quality ratio of cement to the sediment | curing time (days) | number of identical samples | number of replicates |
|---|---|---|---|---|---|
| field rainfall test | UCS | 1 : 4 | 3, 7, 14, 28 | 3 | 1 |
| | TCLP | | 3, 7, 14, 28 | 3 | 3 |
| semi-dynamic leaching test | | 28 | | 1 | 3 |

The experimental procedures are referred to USEPA Method 1315 [24] and ASTM 1308-08 Test [25]. Based on the average pH of 4.46 of rainfall in Shanghai, the SAR was prepared by diluting nitric acid ($HNO_3$) and ammonium sulfate (($NH_4$)$_2SO_4$) in the deionized water in preparation for the extraction leachant in the semi-dynamic leaching test. The stock solutions of SAR were adjusted to three pH values of 2.0, 4.0 and 7.0, respectively, representing strong acidic, weak acidic and neutral conditions.

Under initial pH values of 2.0, 4.0 and 7.0, three sets of parallel experiments were conducted with solidified matrix after 28 days of curing. In the leaching experiment, the volume of leachant was 1121 ml, the surface area of sample was about 118 $cm^2$ and the ratio of sample surface area to volume was 1 : 9.5 ($cm^2$ : ml) according to ASTM 1308-08 [25].

The leachate was collected and the replacement interval was 0.5, 1, 2, 3, 4, 5 and 19 days. Before the replacement of the leachant, HAD-421 was used to measure the leachate pH. The 10 ml of leachate was sampled and filtered with 0.45 μm pore-diameter membrane and acidized with concentrated $HNO_3$ before analysing heavy metals concentration with ICP-MS (Perkin Elmer Nexion 300x). Triplicate samples and blanks were tested to maintain the accuracy and error. The specific experimental conditions and test times are listed in table 4.

# 3. Results and discussion

## 3.1. Effect of rainfall on solidified and stabilized sediment

Rainfall experiments were performed on the specimens, which were cured for 3, 7, 14 and 28 days, respectively. Figure 2 demonstrates that the UCS of specimens after rainfall tests increased with the increased curing time, which was compared with the standard curing conditions as benchmark. The maximum UCS reached 564 kPa under standard curing conditions. After rainfall tests, the maximum UCS of the specimens was 519 kPa. Moreover, the loss of UCS was reduced, which was less affected

**Table 5.** Stabilization rates of leached $Cd^{2+}$ $Pb^{2+}$ and $Cu^{2+}$ with different curing conditions (the percentage indicates the heavy metal stabilization rate after 28 days).

| heavy metal content | Cd | Cu | Pb |
|---|---|---|---|
| the tested sediments (mg kg$^{-1}$) | 1.11 | 60.4 | 156 |
| standard curing condition (mg kg$^{-1}$) | 0.0604 | 12.2 | — |
| stabilization rates (%) | 94.6 | 79.8 | — |
| after rainfall test (mg kg$^{-1}$) | 0.0825 | 13.0 | — |
| stabilization rates (%) | 92.6 | 78.5 | — |

— means the concentration is below detection limit.

by rainfall under longer curing age. The maximum deterioration rate was 34.23% of the UCS after 7 days of curing. The minimum rate was 7.98% after 28 days.

Compared with the results of other publications [8,9,20,26,27], the trend is consistent that UCS increased with increased curing time (1, 7, 28 days in most cases), which is 28 > 14 > 7 > 3 days. The longer curing time resulted in the better strength among different circumstances. Those studies select different UCS values of standard materials as benchmark, such as 1.0 MPa [26], 30 MPa [8,9,20] and 45 MPa [9,20]. Among these values, 30 MPa is mostly used [8,9,20]. However, our study used waste sediments, for which the UCS of 300 kPa was adequate for landfill. The Resource Conservation and Recovery Act (RCRA) of the USA suggests the UCS of solid waste for landfill treatment should be greater than 300 kPa [28].

If the specimen was eroded by acid rain, the hydrogen ion reacted with the hydroxide colloids in the material, resulting in the dissolution of the gelation and the decrease in the pH value. After 3 days of curing, the hydration reaction started to occur, a small quantity of hydroxide colloids was dissolved and generated. When the erosion proceeded, the specimen would release the inner substances through cracks on its surface. Therefore, the compressive strength loss of the specimens was 9.2% after 3 days of curing, higher than standard specimens. After 7 days of curing, the hydration reaction was completed. A large number of $Ca(OH)_2$ hydration products were generated in the structure, but they were dissolved by $H^+$ under rainfall effect. The specimen gradually lost alkalinity and the compressive strength dramatically decreased. Figure 2 indicates that the curing time should be longer (14 > 7 days) after rainfall tests to make the UCS qualified. The UCS of standard and rainfall conditions were preferred at 14 days of curing or longer.

After 28 days of curing, the skeleton was almost solidified and stabilized, and the hydration products from cementing particles formed stable structure. The decrease in the compressive strength was limited. Finally, rainfall was corrosive on the hydroxide colloids on the surface, but the corrosion had no significant impact on the compressive strength. The damage was small after rainfall effect after 28 days.

The decrease in leaching toxicity of heavy metals was determined by the stabilization rate. The stabilization rate was defined in the below equation

$$\sigma = (1 - \frac{C_x}{C}) \times 100\%, \tag{3.1}$$

where $\sigma$ is the stabilization rate of heavy metal (%), $C_x$ is the leaching amount of heavy metal X in per unit mass solidified sample (mg kg$^{-1}$) and $C$ is the total amount of heavy metal X in per unit mass raw sediment (mg kg$^{-1}$).

Table 5 shows the different heavy metals content in pretreated sediments, along with the stabilization rates under standard curing conditions and after rainfall tests. The order of heavy metals' stabilization rates was Pb > Cd > Cu of solidified body under standard curing conditions. The stabilization of Pb was the best among the three metals, for it was not detected in the leaching test after solidification and stabilization. The stabilization rates of Cd and Cu under standard curing condition were 94.6% and 79.8%, respectively. The order of stabilization rates was still Pb > Cd > Cu of solidified body after rainfall experiments. Pb was not detected in the leaching test after rainfall either. The stabilization rates of Cd and Cu after rainfall test were 92.6% and 78.5%, respectively. The stabilization rates of Cd and Cu were decreased by 2% and 1.3%, respectively.

After solidification and stabilization, the raw sediments contained many basic components. Heavy metal ions were converted from cations to insoluble precipitate, complex metal hydroxides, carbonate bound, residual fractions, etc. The solidification and stabilization effect of Pb was the best among these heavy

metals. The morphology of Pb in crystals was relatively stable. Among these five morphologies of Pb, the percentage of residual fraction is highest in the solidified body of 28 days of curing [29]. The Fe–Mn oxides bound decreases, but the exchangeable and carbonate bound increase slightly [30]. Sari *et al.* [31] discovered that the adsorption equation of Pb fits with the Freundlich equation and second-order reaction kinetics of adsorption, indicating that the adsorption process of Pb is an ion exchange process. In the hydration process, $Pb^{2+}$ cations permeated quickly into the crystal structure of hydration product, and then generate coralline crystal with C–S–H. Despite the acid rain, there was little precipitation of metal ions after crystal corrosion. After cement solidification and stabilization, $Cd^{2+}$ existed mainly in residual, followed by carbonate bound and Fe–Mn oxides bound; the organic matter bound and exchangeable forms accounted for the least percentage. After analysis, the carbonate bound and Fe–Mn oxides-bound compounds were susceptible to acid rain and easy to release heavy metals, which was consistent with the reduction of the $Cd^{2+}$ stabilization rate after rainfall tests. Under the strong basic condition, the morphology of $Cu^{2+}$ increased rapidly in exchangeable and carbonate-bound compounds. However, Cu is likely to exist in an organic matter-bound fraction, so that the organic-bound Cu is accounted for the main part of the stabilized sediments. Finally, under the same conditions of adsorbent, time and initial concentration, the order of adsorption rate was Pb > Cd > Cu ions.

The reasons of the low stabilization rate of Cu might be as follows: (i) the corrosion of $H^+$ and $SO_4^{2-}$ onto carbonate and hydration products inside specimen released $Cu^{2+}$ into the water solution; and (ii) in strong basic condition, $Cu^{2+}$ might dissolve into the solution in a complex, for some parts of organic bound Cu decomposed and turned unstable, such as acid-soluble form or exchangeable form. Meanwhile, Jiang *et al.* [32] believed the competition did exist among several heavy metal ions if several heavy metal cations were mixed.

## 3.2. Semi-dynamic leaching experiment

### 3.2.1. Theory

The long-term leachability of heavy metals from cement-stabilized sediments is generally evaluated by the ANS 16.1 model [33]. This model is established based on Fick's diffusion theory and standardized by ANS to evaluate the leaching rate as a function of time. In most cases, the leaching of contaminant from cement-stabilized waste forms follows a diffusion-controlled process [34,35]. In order to evaluate the leaching behaviour of heavy metals incorporated in the S/S treatment sediments, the effective diffusion coefficient ($D_e$) is calculated as follows:

$$D_e = \pi \left[ \frac{(a_n/A_0)^2}{(\Delta_t)_n} \right]^2 \left( \frac{V}{S} \right)^2 T, \qquad (3.2)$$

where $D_e$ is the effective diffusion coefficient ($cm^2 s^{-1}$) $a_n$ is the contaminant loss (mg) during the particular leaching period with subscript $n$, $A_0$ is the initial amount of contaminant existing in the specimen (mg), $V$ is the specimen volume ($cm^3$), $S$ is the surface area of specimen ($cm^2$), $(\Delta_t)_n$ is the duration of the leaching period in seconds, $T$ is the time that elapsed to the middle of the leaching period $n$ (s) and the $T$ can be determined by the below equation

$$T = \left[ \frac{1}{2} (t_n^{1/2} + t_{n-1}^{1/2}) \right]^2, \qquad (3.3)$$

where $t_n$ is the total leaching time of the leaching period ($n$).

The effectiveness of immobilization of the heavy metals in the S/S monolith is evaluated by the leachability index (LX), which is the negative log of observation diffusion coefficient ($-\log D_{eobs}$). According to Environment Canada [16], LX is a performance criterion for the utilization and disposal of S/S-treated waste, which can intuitively reflect the migration of heavy metals in S/S-treated waste. If the LX values are bigger than 9, the treatment process is considered effective and S/S waste is suitable for 'controlled utilization', such as road base, quarry rehabilitation and lagoon closure. When the LX value is between 8 and 9, the S/S-treated waste can be disposed of in sanitary landfills. The S/S waste with an LX value smaller than 8 cannot be disposed of. The LX is defined as follows:

$$LX = \left( \frac{1}{n} \right) \sum_1^n \left[ \log \left( \frac{\beta}{D_e} \right) \right], \qquad (3.4)$$

where $\beta$ is 1 $cm^2 s^{-1}$.

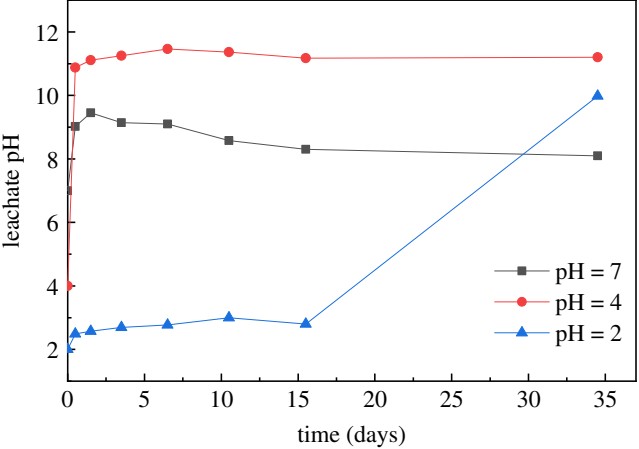

**Figure 3.** Variation of leachate pH during the semi-dynamic leaching tests.

The specific leaching mechanism that controls the release of heavy metals can be determined by the value of the slope of the logarithm of the cumulative release ($\log(B_t)$) versus the logarithm of time ($\log(t)$). When diffusion is the predominant leaching mechanism, the following relationship is established:

$$\log(B_t) = \frac{1}{2}\log(t) + \log\left[U_{\max}d\sqrt{\frac{D_e}{\pi}}\right],$$

(3.5)

where $D_e$ is the effective diffusion coefficient ($m^2\,s^{-1}$) for component $x$ (Cd, Cu and Pb in this study), $B_t$ is the cumulative maximum release of component $x$ ($g\,m^{-2}$), $t$ is the contact time (s), $U_{\max}$ is the maximum leachable quantity ($mg\,kg^{-1}$) and $d$ is the bulk density of S/S product ($kg\,m^{-3}$).

### 3.2.2. Leachate pH value

Figure 3 shows the variation of the leachate pH values with leaching time under different initial pH values of leachant, for the alkaline nature of the S/S matrix affected the initial pH of the leachant significantly. The results showed that alkaline substances were dissolved gradually through the entire leaching process. Generally, the leachate pH increased at the initial stage (0–1.5 days) regardless of the initial pH value. The pH of the leachate under the strong acidic leachant (pH 2.0) was much lower than that under the mildly acidic (pH = 4.0) and neutral (pH = 7.0) leachant during the first 15 days. The leachate pH at pH = 2.0 was below 3.0 during the first 15 days, while the leachate pH for the mildly acidic ranged from about 11.0 to 12.0 and for neutral conditions ranged 8.0–9.5. These variations are attributed to the existence of more $H^+$ in the strong acidic leachant (pH = 2.0) than those hydroxyl ions ($OH^-$) leached from the specimen [36]. The leachate was still acidic. However, the amount of $H^+$ under the mildly acidic (pH = 4.0) and neutral (pH = 7.0) leachant conditions was much less than those $OH^-$ leached from the specimen. Thus, the leachate was basic with pH of 11.0–12.0 and 8.0–9.5 in the early stage.

However, the leachate pH for the case of pH = 2.0 increased significantly to nearly 10.0 after 15 days. Before the 15th day, we replaced the leachant (pH = 2.0) with a short time interval (0.5 and 1 days) and the leachate pH did not change much. The contact and reaction of the acid and the alkaline substances of the module were insufficient, and many alkaline substances in the module were not destroyed. However, after the 15th day, we replaced the leachant with a time interval of 19 days. The long-term effect of strong acid on the specimen made it eroded seriously and many alkaline substances inside leach out gradually [17]. Moreover, more cracks were observed on the module's surface from figure 4, for specimens of leachant pH = 2.0 compared with those of leachant pH = 4.0 and pH = 7.0. These indicated that the strong acid with long-term contact resulted in the increased pH value of leachate.

### 3.2.3. Cumulative fraction of leached heavy metals under different pH conditions

The cumulative fraction of leached heavy metals (CFL) tests below were calculated by the below equation and presented in figure 5

$$CFL = \frac{\sum_i^n c_i \times V_i}{A_0} \times 100\%.$$

(3.6)

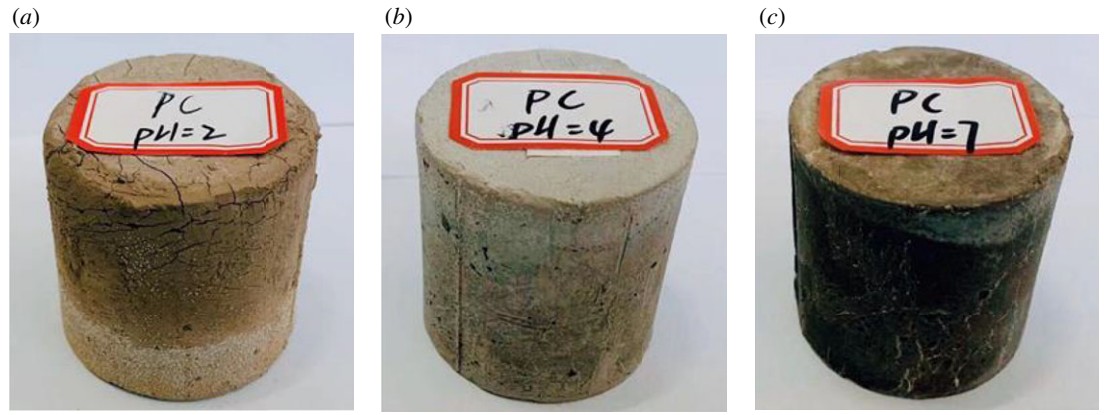

**Figure 4.** Appearance of specimens after leaching 34.5 days: (*a*) pH = 2.0, (*b*) pH = 4.0 and (*c*) pH = 7.0.

**Figure 5.** CFL of target heavy metals (%) under different pH conditions: (*a*) Cu, (*b*) Pb and (*c*) Cd.

where CFL is the cumulative fraction of leached target heavy metals (%), $c_i$ is the concentration of the target heavy metals in the leachate in the leaching period $i$ (mg l$^{-1}$), $V_i$ is the volume of the contact solution (l) and $A_0$ is the initial amount of contaminant present in the specimen (mg).

From figure 5, CFL gradually increased throughout the leaching process. However, the CFL value was less than 2.0% during the overall leaching tests even under the worst case at pH of 2.0, which indicated that the cement products possessed a good stabilization effect on heavy metals. From previous study, chemical fixation of metals in cement occurs via physical or chemical adsorption [37]. Because of the hydration reaction in the cement, heavy metals are adsorbed onto the cementitious products and penetrate the lattices, altering their structure [38]. Hydration products in the S/S materials primarily

**Table 6.** Fitting results for the leaching mechanism determination.

| | pH | slope | $R^2$ | mechanism |
|---|---|---|---|---|
| Cu | 2.0 | 0.54 | 0.972 | diffusion |
| | 4.0 | 0.89 | 0.983 | dissolution |
| | 7.0 | 0.33 | 0.947 | surface wash-off |
| Pb | 2.0 | 0.76 | 0.913 | dissolution |
| | 4.0 | 0.99 | 0.959 | dissolution |
| | 7.0 | — | — | — |
| Cd | 2.0 | 0.83 | 0.966 | dissolution |
| | 4.0 | — | — | — |
| | 7.0 | — | — | — |

consisted of calcium silicate hydrate (C–S–H) and calcium hydroxide Ca(OH)$_2$. These substances either capsulized or precipitated the heavy metals. Therefore, the leachability of heavy metals was reduced.

Furthermore, from figure 5, the leachant pH made significant impacts on the leachability of the heavy metals in the S/S material. Under the strong acidic condition (pH = 2.0), three heavy metals Cu, Pb and Cd were detected, which indicated that the strong acid solution leached out all three heavy metals. However, Cd was undetectable under the mildly acidic condition (pH = 4.0), and neither Pb nor Cd could be detected under neutral condition (pH = 7.0).

From the case of Cu, the CFL at 828 h (34.5 days) was 0.019, 0.16 and 0.64% for the solidified matrix immersed in leachant with a pH of 7.0, 4.0 and 2.0, respectively. The CFL with leachant pH of 2.0 was four times that with leachant pH of 4.0 and about 34 times that with leachant pH of 7.0, respectively. This was probably due to the corrosion under strong acidic condition, so that there were many small cracks on the specimen to release heavy metals. Moreover, it was noted in figure 5 that the CFL curve turned flat gradually after 372 h, which could be attributed to the longer time interval of replacement of leachant solution.

### 3.2.4. The dominant leaching mechanism

Based on the diffusion model developed by Groot & Sloot [39], three main mechanisms were adopted onto the release of heavy metals during the leaching processes: diffusion, dissolution and surface wash-off.

According to USEPA Method 1315 [24], the slope of log($B_t$) is a function of the logarithm of time (log($t$); equation (3.5)). Surface wash-off is the dominant leaching mechanism if the slope is less than or equal to 0.35. If it is close to 0.5 (0.35 < slope ≤ 0.65), the controlling leaching mechanism is diffusion, which is normal in most cement-based materials. The predominant leaching mechanism is defined as dissolution if the slope is greater than 0.65.

The fitting curves were calculated by equation (3.5) in order to illustrate leaching mechanisms under different leaching environments. Meanwhile, all the slopes and $R^2$ values of the fitting curves are presented in table 6.

From table 6, Cu was controlled by different leaching mechanism under different conditions. Its slope value was between 0.35 and 0.65 under strong acidic condition, indicating diffusion as dominant leaching mechanism. Similarly, the controlled leaching mechanism was dissolution under weakly acidic condition and surface wash-off under neutral condition. Generally, dissolution was the controlled leaching mechanism under strong acidic condition. The erosion effect led to an increased surface area exposed, and then accelerated the release and dissolution of Cu$^{2+}$. Meanwhile, diffusion was normally the major release mechanism under weak acidic condition or neutral condition in monolithic materials. However, the leaching phenomenon was a little different for this study, with dissolution at pH = 4. This might be attributed to a large amount of Cu$^{2+}$ released to the soaking solution in the initial immersion period under strong acidic condition. Then, less Cu was migrated from the inner of S/S matrix with a relatively low release rate. Therefore, the slope was bigger than 0.65 at pH = 4.0 group.

Moreover, Pb exhibited dissolution leaching mechanism both in strong and weak acidic solution. This might be attributed to the high concentration of hydrogen ion (H$^+$) and sulfate ion (SO$_4^{2-}$) in soaking solution under low pH value. The ettringite and gypsum might be generated by the external acidified

**Table 7.** Slope values of each leaching period for Cu with leachant pH = 2.0.

| number | leaching period (days) | slope value of log($B_t$) versus log($t$) | $D_e$ (cm$^2$ s$^{-1}$) | LX (average) |
|---|---|---|---|---|
| 1 | 0.5 | — | — | |
| 2 | 1 | 0.68[a] | — | |
| 3 | 2 | 0.54 | $1.17 \times 10^{-11}$ | |
| 4 | 3 | 0.56 | $1.32 \times 10^{-11}$ | 10.89 |
| 5 | 4 | 0.55 | $1.35 \times 10^{-11}$ | |
| 6 | 5 | 0.78[a] | — | |
| 7 | 19 | 0.10[b] | — | |

[a]Dissolution controlled leaching mechanism.
[b]Surface wash-off controlled leaching mechanism.

sulfate attack, based on the reactions below. Therefore, many Pb$^{2+}$ ions were released rapidly, which showed the leaching mechanism was dissolution

$$3CaO \cdot SiO_2 + xH_2O \rightarrow 2CaO \cdot yH_2O + Ca(OH)_2 \rightarrow 2CaO \cdot SiO_2 \cdot mH_2O + 2Ca(OH)_2 \quad (3.7)$$

and

$$3CaO \cdot Al_2O_3 + xH_2O + Ca(OH)_2 \rightarrow 3CaO \cdot Al_2O_3 \cdot mH_2O. \quad (3.8)$$

At last, dissolution was the leaching mechanism for Cd in strong acidic soaking solution, but it was not detected in the solution with pH = 4.0 and pH = 7.0 group. The strong stabilization effect of Cd might be attributed to the adsorption of C–S–H and other hydroxide colloids onto Cd.

### 3.2.5. Long-term effectiveness evaluation of cement-stabilized sediments

The long-term leaching behaviour of heavy metals in S/S materials were assessed with the effective diffusion coefficient ($D_e$), calculated by equation (3.2). In order to interpret the results, the negative log of effective diffusivities ($D_e$) in cm$^2$ s$^{-1}$ was calculated by equation (3.5), which was defined as the LX in equation (3.4).

Before calculating the diffusion coefficient of Cu, it is necessary to discern the leaching mechanism in each leaching period. Then leaching periods with slope values between 0.35 and 0.65 were selected to calculate the diffusion coefficient. The calculated results are shown in table 7.

The average value of $D_e$ for Cu is $1.28 \times 10^{-11}$ cm$^2$ s$^{-1}$. According to Rachana & Rubina [40], the mobility of the contaminants is relatively low if $D_e$ is smaller than $3 \times 10^{-9}$ cm$^2$ s$^{-1}$, while the contaminant is prone to high mobility if $D_e$ is larger than $1 \times 10^{-7}$ cm$^2$ s$^{-1}$. The mobility of contaminants is medium when $D_e$ is between $3 \times 10^{-9}$ cm$^2$ s$^{-1}$ and $1 \times 10^{-7}$ cm$^2$ s$^{-1}$. From table 7, if the mean $D_e$ for Cu was smaller than $3 \times 10^{-1}$ cm$^2$ s$^{-1}$, the mobility was quite low.

Moreover, from table 7, LX for Cu was 10.89 under strong acidic condition, higher than 9. According to Environment Canada [41], the treatment process for Cu was effective. However, it cannot be concluded whether all the S/S treatment sediments could be used under 'controlled utilization' conditions, such as road-base material, quarry rehabilitation, etc. The safety of Pb and Cd in S/S materials awaited further analysis.

### 3.2.6. Another method for calculating effective diffusion coefficient

According to ASTM C1308-08 [25], another method was used to calculate the diffusion coefficient ($D_i$) based on the cumulative fraction of leaching heavy metals equation (3.9).

$$CFL = \frac{\sum_i^n c_i \times V_i}{A_0} = 2\frac{S}{V}\left(\frac{D_i t}{\pi}\right)^{0.5}\frac{S}{V} \quad (3.9)$$

**Table 8.** Diffusion coefficient ($D_i$) by ASTM C1308-08.

|    | pH | slope value of CFL versus $\sqrt{t}$ | $R^2$ | $D_i$ (cm$^2$ s$^{-1}$) | LX = $-\log(D_i)$ |
|----|-----|-------|-------|--------|--------|
| Cu | 2.0 | $3.88 \times 10^{-6}$ | 0.9572 | $8.18 \times 10^{-12}$ | 11.09 |
|    | 4.0 | $1.06 \times 10^{-6}$ | 0.9941 | $6.11 \times 10^{-13}$ | 12.21 |
|    | 7.0 | $1.05 \times 10^{-7}$ | 0.9128 | $5.99 \times 10^{-15}$ | 14.22 |
| Pb | 2.0 | $3.80 \times 10^{-6}$ | 0.9301 | $7.85 \times 10^{-12}$ | 11.11 |
|    | 4.0 | $2.33 \times 10^{-7}$ | 0.8922 | $2.95 \times 10^{-14}$ | 13.53 |
|    | 7.0 | — | — | — | — |
| Cd | 2.0 | $1.17 \times 10^{-5}$ | 0.9386 | $7.44 \times 10^{-11}$ | 10.13 |
|    | 4.0 | — | — | — | — |
|    | 7.0 | — | — | — | — |

and

$$D_i = \frac{\pi}{4}\left(\frac{\mathrm{CFL}}{\sqrt{t}} \cdot \frac{V}{S}\right)^2. \qquad (3.10)$$

The calculated results are presented in table 8.

From table 8, the diffusion coefficient $D_i$ increased with the decrease in pH, reflecting the acidic effect of diffusion. With the increase in pH, the LX value also increased and exhibited consistency. From table 8, all the LX values were bigger than 9, which indicated the treatment process of heavy metals was effective and S/S wastes can be safely used in the areas of road base, quarry rehabilitation and lagoon closure. It was especially evident for Cu whose $D_i$ under strongly acidic condition was three orders of magnitude larger than that under neutral condition. All $D_i$ values were smaller than $3 \times 10^{-9}$ cm$^2$ s$^{-1}$, which indicated that the mobility of all heavy metals was relatively low. Given that the results obtained from USEPA were different from ASTM, the safety of the S/S-treated sediments needs to be discussed specifically after extra experimental verifications.

# 4. Conclusion

(1) The corrosion effects of H$^+$ onto hydroxide colloids and hydration products, along with the infiltration of SO$_4^{2-}$ ions in rainfall, resulted in a maximum deterioration rate of 34.23% of the compressive strength after 7 days of curing, along with minimum rate of 7.98% after 28 days. The UCS was satisfactory for 14 days of curing after rainfall tests.

(2) After rainfall experiments, the stabilization rates changed little through TCLP tests, which indicated that the rainfall had negligible effect on leaching characteristics of heavy metals in S/S sediments.

(3) All the cumulative fractions of leached heavy metals were less than 2.0% under different pH conditions of 2.0, 4.0 and 7.0. This indicated that the S/S-treated sediments with cement exhibited a good stabilization effect during the semi-dynamic tests.

(4) The controlling leaching mechanism of Cu under strongly acidic condition was diffusion according to USEPA Method 1315. The mean diffusion coefficient $D_e$ of Cu was $1.28 \times 10^{-11}$ cm$^2$ s$^{-1}$, and the diffusion coefficient $D_i$ was calculated as $8.18 \times 10^{-12}$ cm$^2$ s$^{-1}$ according to ASTM C1308-08, which was comparable to the result of USEPA Method 1315. The diffusion coefficient $D_i$ of Cd and Pb were calculated as $7.44 \times 10^{-11}$ and $7.85 \times 10^{-12}$ cm$^2$ s$^{-1}$, respectively, based on ASTM C1308-08. Both results demonstrated that the mobility of Cu was quite low under strong acidic condition.

Ethics. We received ethical approval from a 'lab safety and research ethics committee' of University of Shanghai for Science and Technology to carry out our study. This university provided consent in a Research Ethics and code of lab safety and ethics in scientific research and publication. This study did not involve any animals or humans. Our field study was at Chongming island; there was few aquatic animal in our field, except some aquatic plants. Therefore, it is unnecessary to conduct the fieldwork with 'Animal Care Protocol' for this study.

Data accessibility. The datasets supporting this article have been uploaded as part of the electronic supplementary material, which was the raw data of figures 2, 3 and 5, along with experimental procedures.

Authors' contributions. Y.S. made contributions to completing the main part of the experiments with data analysis and interpretation. Y.S. also drafted the manuscript with revised version together with other corresponding authors D.Z., F.L. and H.S. D.Z., a research group leader, participated in the design of the study and drafted the manuscript for the first version. F.L. carried out the statistical analysis of figure 2 and helped to design and complete the field rainfall test. He also provided the research grant (YRWEF201603). H.T. helped to analyse and interpreted the data of the semi-dynamic leaching tests and completed the linear regression of data figures 3 and 5 and tables 6–8. She also provided the research grant (NSFC 51679140). The student Z.G. and M.L. prepared the specimens and collected field data of the unconfined compressive strength (UCS) test and the toxicity characteristic leaching procedure (TCLP) tests. They were responsible of acquisition and organization of data for tables 1–5 and figure 4. L.M. participated in the design of experimental process of figure 1; she helped the organization of raw data in electronic supplementary material. She also provided research grant (NSFC 41601229). Z.L. coordinated the study and helped Y.S. with the preparation of the sediment samples, and he participated in data analysis of dissociation constant and LX. H.S. completed several intensive revisions and submitted the manuscript repeatedly, he provided research grant (NSFC 21606150). All authors gave final approval for publication.

Competing interests. There are no competing interests in this study.

Funding. This research was financially supported by the National Natural Science Foundation of China (NSFC grant nos 41601229, 51679140, 21606150). This work was also supported by the key laboratory of Yangtze River water environment, Ministry of Education of China (grant no. YRWEF201603).

Acknowledgements. We also thank Yongzheng Li, Yang Yang and Jiao Chen for the completion of original experiments, which did not meet the authorship criteria.

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
