## [Reviewer comments · Royal Society Open Science]

Review History

RSOS-190998.R0 (Original submission)

Review form: Reviewer 1

Is the manuscript scientifically sound in its present form?

No

Are the interpretations and conclusions justified by the results?

No

Is the language acceptable?

Yes

Do you have any ethical concerns with this paper?

No

Have you any concerns about statistical analyses in this paper?

No

Recommendation?

Major revision is needed (please make suggestions in comments)

Comments to the Author(s)

The manuscript presents the study of a river sediment, artificially contaminated, the treated with Portland cement, and the potential of stabilization of Cu, Cd and Pb.

The experimental approach is interesting but there are some missing informations to help at the complete comprehension of the involved stabilization processes.

All the comments are noted on the manuscript (see Appendix A), but some comments are given below.

It is necessary to precise :

- the physical parameters of the river sediment :
 - o PSD : no particles under 5 μm ? no clayey particles (0/2 μm)? So, the contamination is not associated to clay fraction
 - o Atterberg limits : liquidity limit and index plasticity are important to define the clay activity of the sediment, defined as a silty clay
 - o organic matter content if possible, the organic fraction being defined as important in the stabilization processes
- the sampling method of the river sediment (weight, mode...)
- as soon as possible, that the contamination of the sediment is artificial: how the addition concentrations are defined? what it means (contamination, pollution, waste, reuse...)?
- only Cu, Cd and Pb are studied: why not Cr, Ni and Zn, which are also analyzed ?
- the Portland cement 42.5 content used (by weight of dry sediment)
- the objective of the solidification: UCS28 = 0.76 MPa is weak for subgrade layer for example
- the Portland cement content in Cu, Cd and Pb: Cu in an usual element of the cements and it is important to determine is addition
- the number of samples/specimens by test: then the figures have to present the standard deviations or uncertainties of the measurements
- as soon as possible that the mechanical performance measured is unconfined compressive strength (UCS) (not hydraulic properties)
- the dimensions of the samples on which UCS tests are realized: it is recommended to realize UCS test on samples of slenderness 2 (high/diameter)
- the references for some affirmation (noted "REF?") or the standards used for samples confection, conservation, the SAR, etc.
- in conclusion, first, the materials used for the experimental study, before the synthesis of the results

Moreover it is necessary to explain/ discuss of:

- the impact of a decrease of 8% of the UCS after acid rainfall: what are the technical applications?
- the impact of a weak stabilization: what are the technical (chemical) referential? Environmental thresholds? what is the interest of the diffusion equations with the technical applications?
- are specific form of erosion observed on the samples/specimens surface? does the leaching reached the sample/specimen core (sufficient porosity or permeability)?
- with the obtained results, the intact (micro)structure and the stability of the cement treated sediment

The use of "stabilization" term to S/S sediment supposes that the stabilization is effective.

The gypsum $\text{CaSO}_4 \cdot 2\text{H}_2\text{O}$ is not an expansive mineral; ettringite is.

If gypsum is formed after rainfall on the treated sediment after 3 days of cure, it could decrease the porosity. Is it observed by DRX or SEM/EDS-EDX?

The gypsum is not an hydrate which can binded the soil particles.

The processes of dissolution of the minerals lead to recombination in new minerals but not in total destruction.

Some references to analyse the results are not adapted to the context. For example, it is probably not relevant to compare the geochemical behavior of a kaolinite clay to that of a silty clay.

On the form:

- use the present form if possible, rather than the past form
- take care of the precision of the data
- take care of inconsistencies between the abstract and the manuscript or the submission: keywords of the presentation page are different of keywords of the abstract; samples/specimens subjected to 3 days of cure; presentation of Zn as toxic metal (metalloïd) in introduction, not studied after...
- replace:
 - o "KPa" by "kPa"
 - o "PH" by "pH"
 - o "crystals" by "minerals"
- see manuscript annotations

Review form: Reviewer 2

Is the manuscript scientifically sound in its present form?

No

Are the interpretations and conclusions justified by the results?

No

Is the language acceptable?

No

Do you have any ethical concerns with this paper?

No

Have you any concerns about statistical analyses in this paper?

No

Recommendation?

Reject

Comments to the Author(s)

Review of RSOS-190998 "The rainfall effect onto solidification and stabilization of heavy metal-polluted sediments"

General comments:

This study aims to investigate the effect of acid rain on hydraulic properties and leaching characteristics of cement-based solidified sediments.

The introduction of the manuscript should be reshaped. And it is important to introduce how the research is built on relevant studies and what is the research objective.

As to my knowledge, the effect of acid rain on cement-based solidified sediments is widely documented in literatures. The novelty and academic merits of this study are not properly

justified. The structures and contents of the manuscript should be further improved. In addition, the present form does not have sufficient results presenting the novelty of a high-quality journal paper.

I believe the manuscript should be rejected. Specific comments are provided as follows:

Specific comments:

Introduction:

Many references cited in the manuscript are more than 10 years ago. Thus, it does not properly represent advances in scientific knowledge. Recent papers about sediment in the construction industry in 2015-2019 should be critically reviewed (e.g., *Cement Concr. Compos.* 2019, 104, 103348; *J. Clean. Prod.*, 2018, 199, 69-76) in the manuscript.

Materials & methods:

Page 4: Please provide the cement ratio in the solidified samples. This is important to the performance of the solidified samples.

Did the authors consider the influence of organic matter in the sediment on the cement hydration?

Discussion:

Page 12-13: There are so many paragraphs in this section just simply discuss the experimental results without solid evidence and scientific explanation.

Tables, figures should be improved. Key results are not well presented and it is very hard to follow.

Decision letter (RSOS-190998.R0)

03-Oct-2019

Dear Dr Shi:

Manuscript ID RSOS-190998 entitled "The rainfall effect onto solidification and stabilization of heavy metal-polluted sediments" which you submitted to Royal Society Open Science, has been reviewed. The comments from reviewers are included at the bottom of this letter.

In view of the criticisms of the reviewers, the manuscript has been rejected in its current form. However, a new manuscript may be submitted which takes into consideration these comments.

Please note that resubmitting your manuscript does not guarantee eventual acceptance, and that your resubmission will be subject to peer review before a decision is made.

Your resubmitted manuscript should be submitted by 01-Apr-2020. If you are unable to submit by this date please contact the Editorial Office.

Kind regards,
Anita Kristiansen
Royal Society Open Science Editorial Office
Royal Society Open Science
openscience@royalsociety.org

on behalf of Dr Riccardo Avanzinelli (Associate Editor) and Jon Blundy (Subject Editor)
openscience@royalsociety.org

Associate Editor Comments to Author (Dr Riccardo Avanzinelli):

Associate Editor:

Comments to the Author:

Dear Dr Huancong,

Your manuscript has been examined by two experts who provided important criticisms and overall negative reviews, recommending major revision and outright rejection, respectively. Reviewer #1 found the experimental approach interesting, but found several major issues including i) the lack of several important details and information on the experiments that were not provided in the manuscript, and ii) the insufficient discussion and explanation of some important aspects of the experiments and their implications. Reviewer #2 rejected the paper stating that it failed to reference several recent studies made on the same subjects and stating the discussion section is made without solid evidence and scientific explanation.

From my own reading of the paper, I agree with the reviewers criticisms, and I also found other major issues, especially regarding the description of the analytical methods used in this study. The manuscript focuses on the mobility of toxic elements (Cu, Pb, Cd) in sediments after stabilization with cement-based binder and most of the discussion is based on measurements of the concentration of these elements in the samples before and after the experiments. However no indications are provided on the procedures or even the analytical methods used to determine such concentrations, and the resulting analytical errors.

Therefore, according to these reviews and from my own evaluation, I believe the manuscript has too many shortcomings and it does not meet the standard for publication on RSOS. My recommendation is "Reject and allow resubmission", considering that the experimental approach is interesting (as stated also by reviewer #1) and that the study has the potential to provide a useful contribution, but only if the manuscript is completely restructured and significantly implemented taking into consideration the reviewers' comments and criticism.

Reviewers' Comments to Author:

Reviewer: 1

Comments to the Author(s)

The manuscript presents the study of a river sediment, artificially contaminated, the treated with Portland cement, and the potential of stabilization of Cu, Cd and Pb. The experimental approach is interesting but there are some missing informations to help at the complete comprehension of the involved stabilization processes.

All the comments are noted on the manuscript, but some comments are given below.

It is necessary to precise :

- the physical parameters of the river sediment :

- o PSD : no particles under 5 μm ? no clayey particles (0/2 μm)? So, the contamination is not associated to clay fraction
- o Atterberg limits : liquidity limit and index plasticity are important to define the clay activity of the sediment, defined as a silty clay
- o organic matter content if possible, the organic fraction being defined as important in the stabilization processes
 - the sampling method of the river sediment (weight, mode...)
 - as soon as possible, that the contamination of the sediment is artificial: how the addition concentrations are defined? what it means (contamination, pollution, waste, reuse...)?
 - only Cu, Cd and Pb are studied: why not Cr, Ni and Zn, which are also analyzed ?
 - the Portland cement 42.5 content used (by weight of dry sediment)
 - the objective of the solidification: UCS28 = 0.76 MPa is weak for subgrade layer for example
 - the Portland cement content in Cu, Cd and Pb: Cu in an usual element of the cements and it is important to determine is addition
 - the number of samples/specimens by test: then the figures have to present the standard deviations or uncertainties of the measurements
 - as soon as possible that the mechanical performance measured is unconfined compressive strength (UCS) (not hydraulic properties)
 - the dimensions of the samples on which UCS tests are realized: it is recommended to realize UCS test on samples of slenderness 2 (high/diameter)
 - the references for some affirmation (noted "REF?") or the standards used for samples confection, conservation, the SAR, etc.
 - in conclusion, first, the materials used for the experimental study, before the synthesis of the results

Moreover it is necessary to explain/ discuss of:

- the impact of a decrease of 8% of the UCS after acid rainfall: what are the technical applications?
- the impact of a weak stabilization: what are the technical (chemical) referential? Environmental thresholds? what is the interest of the diffusion equations with the technical applications?
- are specific form of erosion observed on the samples/specimens surface? does the leaching reached the sample/specimen core (sufficient porosity or permeability)?
- with the obtained results, the intact (micro)structure and the stability of the cement treated sediment

The use of "stabilization" term to S/S sediment supposes that the stabilization is effective.

The gypsum $\text{CaSO}_4 \cdot 2\text{H}_2\text{O}$ is not an expansive mineral; ettringite is.

If gypsum is formed after rainfall on the treated sediment after 3 days of cure, it could decrease the porosity. Is it observed by DRX or SEM/EDS-EDX?

The gypsum is not an hydrate which can binded the soil particles.

The processes of dissolution of the minerals lead to recombination in new minerals but not in total destruction.

Some references to analyse the results are not adapted to the context. For example, it is probably not relevant to compare the geochemical behavior of a kaolinite clay to that of a silty clay.

On the form:

- use the present form if possible, rather than the past form
- take care of the precision of the data
- take care of inconsistencies between the abstract and the manuscript or the submission: keywords of the presentation page are different of keywords of the abstract; samples/specimens

subjected to 3 days of cure; presentation of Zn as toxic metal (metalloid) in introduction, not studied after...

- replace:
 - o "KPa" by "kPa"
 - o "PH" by "pH"
 - o "crystals" by "minerals"
- see manuscript annotations

Reviewer: 2

Comments to the Author(s)

Review of RSOS-190998 "The rainfall effect onto solidification and stabilization of heavy metal-polluted sediments"

General comments:

This study aims to investigate the effect of acid rain on hydraulic properties and leaching characteristics of cement-based solidified sediments.

The introduction of the manuscript should be reshaped. And it is important to introduce how the research is built on relevant studies and what is the research objective.

As to my knowledge, the effect of acid rain on cement-based solidified sediments is widely documented in literatures. The novelty and academic merits of this study are not properly justified. The structures and contents of the manuscript should be further improved. In addition, the present form does not have sufficient results presenting the novelty of a high-quality journal paper.

I believe the manuscript should be rejected. Specific comments are provided as follows:

Specific comments:

Introduction:

Many references cited in the manuscript are more than 10 years ago. Thus, it does not properly represent advances in scientific knowledge. Recent papers about sediment in the construction industry in 2015-2019 should be critically reviewed (e.g., *Cement Concr. Compos.* 2019, 104, 103348; *J. Clean. Prod.*, 2018, 199, 69-76) in the manuscript.

Materials & methods:

Page 4: Please provide the cement ratio in the solidified samples. This is important to the performance of the solidified samples.

Did the authors consider the influence of organic matter in the sediment on the cement hydration?

Discussion:

Page 12-13: There are so many paragraphs in this section just simply discuss the experimental results without solid evidence and scientific explanation.

Tables, figures should be improved. Key results are not well presented and it is very hard to follow.

Author's Response to Decision Letter for (RSOS-190998.R0)

See Appendix B.

RSOS-192234.R0

Review form: Reviewer 1

Is the manuscript scientifically sound in its present form?

No

Are the interpretations and conclusions justified by the results?

Yes

Is the language acceptable?

Yes

Do you have any ethical concerns with this paper?

No

Have you any concerns about statistical analyses in this paper?

No

Recommendation?

Major revision is needed (please make suggestions in comments)

Comments to the Author(s)

The revised manuscript is clearly improved, but some surprising lacks are encountered (more than typographic checking).

For example, in abstract and conclusion paragraph, only the Cu behaviour is presented, and not on Pb and Cd.

Other example, the used assumptions shall be carefully documented : for example, the use of nitrate solutions to contaminate the sediments can delay the setting and the hardening of the cement then of the treated sediments, and influence the stabilisation of the heavy metals.

Last example, the responses to the reviewer should be included in the manuscript. If the mechanical performance of the treated sediments decrease of 14%, what is the damaging effect on site ?

In conclusion, the manuscript needs major revision.

Review form: Reviewer 2

Is the manuscript scientifically sound in its present form?

No

Are the interpretations and conclusions justified by the results?

No

Is the language acceptable?

Yes

Do you have any ethical concerns with this paper?

No

Have you any concerns about statistical analyses in this paper?

Yes

Recommendation?

Major revision is needed (please make suggestions in comments)

Comments to the Author(s)

The authors made some revisions to the manuscript. However, some critical issues still remain unaddressed. The novelty and academic merits of this study are not properly justified. Specific comments are provided as follows:

The authors just simply neglected the suggested references. The recent papers on solidification and stabilization of contaminated sediment have to be critically reviewed and compared in this study, otherwise this study is clearly not built on the latest findings in the existing literature and cannot be accepted for publication. For example:

The roles of biochar as green admixture for sediment-based construction products. *Cem. Concr. Compos.* 104 (2019) 103348

Novel synergy of Si-rich minerals and reactive MgO for stabilisation/solidification of contaminated sediment. *J. Hazard. Mater.* 365 (2019) 695-706

Recycling dredged sediment into fill materials, partition blocks, and paving blocks: Technical and economic assessment. *J. Clean. Prod.* 199 (2018) 69-76

Green remediation of contaminated sediment by stabilization/solidification with industrial by-products and CO₂ utilization. *Sci. Total Environ.* 631 (2018) 1321-1327

Recycling contaminated sediment into eco-friendly paving blocks by a combination of binary cement and carbon dioxide curing. *J. Clean. Prod.* 164 (2017) 1279-1288

Fig. 2 What is the requirement of compressive strength for these products?

Line 46-48: The authors said gypsum and ettringite were generated. Please provide direct evidence.

The experimental results are simply described without in-depth discussion based on scientific understanding.

Nevertheless, the manuscript can still provide the scientific community with some useful contribution to improving knowledge of solidification and stabilization of contaminated sediment. I recommend major revision.

Decision letter (RSOS-192234.R0)

27-Mar-2020

Dear Dr Shi,

The Subject Editor assigned to your paper ("The rainfall effect onto solidification and stabilization of heavy metal-polluted sediments") has now received comments from reviewers. We would like you to revise your paper in accordance with the referee and Associate Editor suggestions which can be found below (not including confidential reports to the Editor). Please note this decision does not guarantee eventual acceptance.

It is clear that some modifications to the original submission have been made, but neither reviewers nor Associate Editor were convinced by your efforts, hence the recommendation for Major Revision. Please consider carefully the comments and take care in making your revisions. There is a suggestion that you rather hurried the revisions this time around, and that this has impacted on the quality of your manuscript. As this is a resubmission of your original manuscript, this is your final chance to make the required changes. There can be no further rounds of Major Revision to get the article into shape - if we do not feel that the revised version reflects adequately the reviewers' recommendation, your manuscript will be rejected, I am afraid.

Please submit a copy of your revised paper before 19-Apr-2020. Please note that the revision deadline will expire at 00.00am on this date. If we do not hear from you within this time then it will be assumed that the paper has been withdrawn. In exceptional circumstances, extensions may be possible if agreed with the Editorial Office in advance. We do not allow multiple rounds of revision so we urge you to make every effort to fully address all of the comments at this stage. If deemed necessary by the Editors, your manuscript will be sent back to one or more of the original reviewers for assessment. If the original reviewers are not available we may invite new reviewers.

When submitting your revised manuscript, you must respond to the comments made by the referees and upload a file "Response to Referees" in "Section 6 - File Upload". Please use this to document how you have responded to each of the comments, and the adjustments you have made. In order to expedite the processing of the revised manuscript, please be as specific as possible in your response.

- Ethics statement

- Data accessibility

If you wish to submit your supporting data or code to Dryad (<http://datadryad.org/>), or modify your current submission to dryad, please use the following link:
<http://datadryad.org/submit?journalID=RSOS&manu=RSOS-192234>

- Competing interests

- Authors' contributions

- Acknowledgements

- Funding statement

Additionally, please update the following email address when resubmitting:

ziyang.ling@student.manchester.ac.uk

Kind regards,

Lianne Parkhouse
Royal Society Open Science
openscience@royalsociety.org

on behalf of Dr Riccardo Avanzinelli (Associate Editor) and Jon Blundy (Subject Editor)
openscience@royalsociety.org

Associate Editor Comments to Author (Dr Riccardo Avanzinelli):

Dear Dr. Shi,

The re-submitted version of your manuscript has been evaluated by two reviewers. Both reviewers acknowledged the substantial work made by the authors, but still found that major revisions are still required.

Reviewer#1, stated that the novelty and academic merits of this study are not properly justified and that you did not consider some of the previous works and key references that were suggested in the previous round of review. This reviewer also found that the experimental results are not discussed enough.

Reviewer #2 found that failed to define the application of the experimental study in the manuscript, and spotted several basic errors in the text.

From my own reading of the manuscript I also found the manuscript significantly improved with respect to the previous version, but I agree with the reviewers that it still requires several significant improvements. The discussion and explanation of some of the data is not provided with enough evidence; for example, as highlighted by reviewer #1, the role of gypsum and ettringite is proposed without providing any direct evidence of the presence of these mineral phases on the studied samples.

Also invite you to carefully check the language and especially the Tables. The data reported in the Tables are often difficult to link to what discussed in the text. For example:

1) Table 4 reports no stabilisation rate (contrary to what stated in its caption). I tried to calculate the stabilisation rates using the formula provided in the text, but the values came out different to those reported in the text. I also found the definition of the sigma value in the text confusing (e.g., Cx should be a concentration and not a total amount and I guess it is “the concentration of metal X in the leachate of the solidified body” and not “the leaching amount of heavy metal X in solidified body(mg·kg-1)”.

2) In Table 6 and 7 the De and i values are reported in cm²/sec whilst in the text are in m²/sec. That is quite confusing and make it difficult to follow the reasoning of the discussion, please homogenise.

3) LX is reported in Table 6 and not in Table 5 (Page 19, line 23).

Therefore, according to the reviewers' comments and from my own evaluation, I believe the manuscript require Major Revision in order to be accepted for publication on Royal Society Open Science.

Reviewer comments to Author:

Reviewer: 1
Comments to the Author(s)

The revised manuscript is clearly improved, but some suprising lacks are encountered (more than typographic checking).

For example, in abstract and conclusion paragraph, only the Cu behaviour is presented, and not on Pb and Cd.

Other example, the used assumptions shall be carefully documented : for example, the use of nitrate solutions to contaminate the sediments can delay the setting and the hardening of the cement then of the treated sediments, and influence the stabilisation of the heavy metals.

Last example, the responses to the reviewer should be included in the manuscript. If the mechanical performance of the treated sediments decrease of 14%, what is the damaging effect on site ?

In conclusion, the manuscript needs major revision.

Reviewer: 2
Comments to the Author(s)

The authors made some revisions to the manuscript. However, some critical issues still remain

unaddressed. The novelty and academic merits of this study are not properly justified. Specific comments are provided as follows:

The authors just simply neglected the suggested references. The recent papers on solidification and stabilization of contaminated sediment have to be critically reviewed and compared in this study, otherwise this study is clearly not built on the latest findings in the existing literature and cannot be accepted for publication. For example:

The roles of biochar as green admixture for sediment-based construction products. *Cem. Concr. Compos.* 104 (2019) 103348

Novel synergy of Si-rich minerals and reactive MgO for stabilisation/solidification of contaminated sediment. *J. Hazard. Mater.* 365 (2019) 695-706

Recycling dredged sediment into fill materials, partition blocks, and paving blocks: Technical and economic assessment. *J. Clean. Prod.* 199 (2018) 69-76

Green remediation of contaminated sediment by stabilization/solidification with industrial by-products and CO₂ utilization. *Sci. Total Environ.* 631 (2018) 1321-1327

Recycling contaminated sediment into eco-friendly paving blocks by a combination of binary cement and carbon dioxide curing. *J. Clean. Prod.* 164 (2017) 1279-1288

Fig. 2 What is the requirement of compressive strength for these products?

Line 46-48: The authors said gypsum and ettringite were generated. Please provide direct evidence.

The experimental results are simply described without in-depth discussion based on scientific understanding.

Nevertheless, the manuscript can still provide the scientific community with some useful contribution to improving knowledge of solidification and stabilization of contaminated sediment. I recommend major revision.

Author's Response to Decision Letter for (RSOS-192234.R0)

See Appendix C.

RSOS-192234.R1 (Revision)

Review form: Reviewer 1

Is the manuscript scientifically sound in its present form?

Yes

Are the interpretations and conclusions justified by the results?

Yes

Is the language acceptable?

Yes

Do you have any ethical concerns with this paper?

No

Have you any concerns about statistical analyses in this paper?

No

Recommendation?

Major revision is needed (please make suggestions in comments)

Comments to the Author(s)

The authors made some revisions to the manuscript. Some others could improve its quality, based on previous observations and responses.

The carbonation is a pathology of the cementitious materials. It cannot be identify as an S/S process improvement.

It is necessary to precise that the samples contain only sand and powder, but little clay minerals and secondary minerals. The experimental sediment consists of 3.7% clay particles, 70.9% silt particles, 25.4% sand particles in Table 1 for example. So, the sediment is defined as a silt.

The addition concentration is based on China's specification "GB 15618-2018. Soil environment quality risk control standard for soil contamination of agriculture land".

The reference to justify nitrate solution addition is not relevant.

Only Cu, Cd and Pb are studied: why not Cr, Ni and Zn, which are also analyzed ? The answer is to insert in the manuscript.

In the manuscript, to add how much samples are tested for each configuration of test (UCS and leaching tests), what is the repetability of the tests : we prepared three groups of parallel samples and took the average values. The error bars were shown in the chart of Figure 2 and 5.

In abstract, introduction and conclusion, to indicate the objective of reuse of the sediments : landfill.

According to the Chinese design code "regulations in the design code for highway subgrade (JTG d30-2015)", the UCS index of the lightweight soil of the subgrade should be greater than 600 kPa for expressways and first-class highways, and more than 500 kPa for secondary and sub-secondary highways. The UCS of the solidified sediment meets the specification requirements, so the heavy metal contaminated sediment can be used as roadbed filler after solidified and stabilized.

Koenig A [1] suggest that the UCS of waste should be greater than 50 kPa for landfill treatment, while the resource conservation and recovery act (RCRA) of the United States suggests that it should be greater than 300 kPa [2]. In this study, the recommended value of RCRA is 300 kPa, so the curing energy can meet the requirement of building backfill strength.

[1] Koenig A, Kay J N, Wan I M. Physical properties of dewater sludge for landfilling [J]. Water Science and Technology, 1996, 34:533-540.

[2] Shi C, Fernández-Jiménez A. Stabilization/solidification of hazardous and radioactive wastes with alkali-activated cements [J]. Journal of hazardous materials, 2006, 137(3): 1656-1663.

To precise that, taking the strength qu_0 at the ratio of height to diameter of 2 as the reference value, through regression analysis, the ratio of qu to qu_0 at any ratio of height to diameter of H/D is obtained as follows:

$$qu / quo = 0.983(H/2D)-0.455$$

To indicate "Malviya and Chaudhary pointed out that when the observation diffusion coefficient $Di_{obs} < 3 \times 10^{-13} \text{ m}^2/\text{s}$, the pollutant migration is low. When $3 \times 10^{-13} \text{ m}^2/\text{s} < Di_{obs} < 10^{-11} \text{ m}^2/\text{s}$, the pollutant migration is moderate. When $Di_{obs} > 10^{-11} \text{ m}^2/\text{s}$, pollutant migration is high. In order to interpret the results, the negative log of observation diffusion coefficient ($-\lg Di$

obs) can be taken to define the Leachability index of heavy metals after correction. Environment Canada has listed this parameter as a control index for the utilization and disposal of solidified and stabilized contaminated soils. The formula for LX is shown below. When LX is greater than 9, it indicates that the curing and stabilization treatment of contaminated soil is effective and can be used for controlling the curing stop, such as as roadbed filler. When LX is greater than 8, the solidified soil can be disposed of by landfill. When LX is less than 8, the solidified soil is not suitable for landfill disposal."

Are specific form of erosion observed on the samples/specimens surface? does the leaching reached the sample/specimen core (sufficient porosity or permeability)?

All of the previous studies suggest that diffusion is the controlling mechanism of most heavy metals that are leaching from S/S treated sediments including this research [1] ,which is not involved in the specimen core.

[1] Yan-Jun Du,Ming-Li Wei,Krishna R. Reddy,Zhao-Peng Liu,Fei Jin. Effect of acid rain pH on leaching behavior of cement stabilized lead-contaminated soil[J]. Journal of Hazardous Materials,2014,271.

Review form: Reviewer 2

Is the manuscript scientifically sound in its present form?

Yes

Are the interpretations and conclusions justified by the results?

Yes

Is the language acceptable?

Yes

Do you have any ethical concerns with this paper?

No

Have you any concerns about statistical analyses in this paper?

Yes

Recommendation?

Accept as is

Comments to the Author(s)

The authors have adequately addressed the comments and improved the manuscript for publication, cheers

Decision letter (RSOS-192234.R1)

Dear Dr Shi:

On behalf of the Editors, I am pleased to inform you that your Manuscript RSOS-192234.R1 entitled "The rainfall effect onto solidification and stabilization of heavy metal-polluted sediments" has been accepted for publication in Royal Society Open Science subject to minor revision in accordance with the referee suggestions. Please find the referees' comments at the end of this email.

The reviewers and Subject Editor have recommended publication, but also suggest some minor revisions to your manuscript. Therefore, I invite you to respond to the comments and revise your manuscript.

- Ethics statement

- Data accessibility

If you wish to submit your supporting data or code to Dryad (<http://datadryad.org/>), or modify your current submission to dryad, please use the following link:
<http://datadryad.org/submit?journalID=RSOS&manu=RSOS-192234.R1>

- Competing interests

- Authors' contributions

- Acknowledgements

- Funding statement

Because the schedule for publication is very tight, it is a condition of publication that you submit the revised version of your manuscript before 29-May-2020. Please note that the revision deadline will expire at 00.00am on this date. If you do not think you will be able to meet this date please let me know immediately.

on behalf of Dr Riccardo Avanzinelli (Associate Editor)
openscience@royalsociety.org

Associate Editor Comments to Author (Dr Riccardo Avanzinelli):

Comments to the Author:
Dear Dr. Shi,

The revised version of your manuscript has been evaluated by two reviewers. Both reviewers acknowledged that you addressed most of the comments and criticism and that the manuscript is substantially improved, but yet the two revisions are slightly contrasting.

Reviewer#1, recommended acceptance as it is, whilst reviewer #2 suggested that major revision are still needed.

From my own reading of the manuscript I also found the manuscript is significantly improved and that you did a good job in addressing the reviewers (and my own) comments. The comments of reviewer # are valid should be taken into account to in order to further improve the manuscript. In fact I think the implementations suggested by reviewer #2 are not too difficult to apply.

Therefore I believe that the manuscript can to be accepted for publication on Royal Society Open Science after minor revisions specifically addressing the points highlighted by reviewer #2. .

Best regards
Riccardo Avanzinelli

Reviewer comments to Author:
Reviewer: 2

Comments to the Author(s)
The authors have adequately addressed the comments and improved the manuscript for publication, cheers

Reviewer: 1

Comments to the Author(s)
The authors made some revisions to the manuscript. Some others could improve its quality, based on previous observations and responses.

The carbonation is a pathology of the cementitious materials. It cannot be identify as an S/S process improvement.

It is necessary to precise that the samples contain only sand and powder, but little clay minerals and secondary minerals. The experimental sediment consists of 3.7% clay particles,70.9% silt particles, 25.4% sand particles in Table 1 for example. So, the sediment is defined as a silt.

The addition concentration is based on China's specification "GB 15618-2018. Soil environment quality risk control standard for soil contamination of agriculture land".

The reference to justify nitrate solution addition is not relevant.

Only Cu, Cd and Pb are studied: why not Cr, Ni and Zn, which are also analyzed ? The answer is to insert in the manuscript.

In the manuscript, to add how much samples are tested for each configuration of test (UCS and leaching tests), what is the repeatability of the tests : we prepared three groups of parallel samples and took the average values. The error bars were shown in the chart of Figure 2 and 5.

In abstract, introduction and conclusion, to indicate the objective of reuse of the sediments : landfill.

According to the Chinese design code "regulations in the design code for highway subgrade (JTG d30-2015)", the UCS index of the lightweight soil of the subgrade should be greater than 600 kPa for expressways and first-class highways, and more than 500 kPa for secondary and sub-secondary highways. The UCS of the solidified sediment meets the specification requirements, so the heavy metal contaminated sediment can be used as roadbed filler after solidified and stabilized.

Koenig A [1] suggest that the UCS of waste should be greater than 50 kPa for landfill treatment, while the resource conservation and recovery act (RCRA) of the United States suggests that it should be greater than 300 kPa [2]. In this study, the recommended value of RCRA is 300 kPa, so the curing energy can meet the requirement of building backfill strength.

[1] Koenig A, Kay J N, Wan I M. Physical properties of dewater sludge for landfilling [J]. Water Science and Technology, 1996, 34:533-540.

[2] Shi C, Fernández-Jiménez A. Stabilization/solidification of hazardous and radioactive wastes with alkali-activated cements [J]. Journal of hazardous materials, 2006, 137(3): 1656-1663.

To precise that, taking the strength qu_0 at the ratio of height to diameter of 2 as the reference value, through regression analysis, the ratio of qu to qu_0 at any ratio of height to diameter of H/D is obtained as follows:

$$qu / qu_0 = 0.983(H/2D)-0.455$$

To indicate "Malviya and Chaudhary pointed out that when the observation diffusion coefficient D_i

$obs < 3 \times 10^{-13} \text{ m}^2/\text{s}$, the pollutant migration is low. When $3 \times 10^{-13} \text{ m}^2/\text{s} < D_i < 10^{-11} \text{ m}^2/\text{s}$, the pollutant migration is moderate. When $D_i > 10^{-11} \text{ m}^2/\text{s}$, pollutant migration is high.

In order to interpret the results, the negative log of observation diffusion coefficient ($-\lg D_i \text{ obs}$) can be taken to define the Leachability index of heavy metals after correction. Environment Canada has listed this parameter as a control index for the utilization and disposal of solidified and stabilized contaminated soils. The formula for LX is shown below. When LX is greater than 9, it indicates that the curing and stabilization treatment of contaminated soil is effective and can be used for controlling the curing stop, such as as roadbed filler. When LX is greater than 8, the solidified soil can be disposed of by landfill. When LX is less than 8, the solidified soil is not suitable for landfill disposal."

Are specific form of erosion observed on the samples/specimens surface? does the leaching reached the sample/specimen core (sufficient porosity or permeability)?

All of the previous studies suggest that diffusion is the controlling mechanism of most heavy metals that are leaching from S/S treated sediments including this research [1], which is not involved in the specimen core.

[1] Yan-Jun Du, Ming-Li Wei, Krishna R. Reddy, Zhao-Peng Liu, Fei Jin. Effect of acid rain pH on leaching behavior of cement stabilized lead-contaminated soil [J]. Journal of Hazardous Materials, 2014, 271.

Author's Response to Decision Letter for (RSOS-192234.R1)

See Appendix D.

Decision letter (RSOS-192234.R2)

Dear Dr Shi:

On behalf of the Editors, I am pleased to inform you that your Manuscript RSOS-192234.R2 entitled "The rainfall effect onto solidification and stabilization of heavy metal-polluted sediments" has been accepted for publication in Royal Society Open Science subject to minor revision in accordance with the referee suggestions. Please find the referees' comments at the end of this email.

The reviewers and Subject Editor have recommended publication, but also suggest some minor revisions to your manuscript. Therefore, I invite you to respond to the comments and revise your manuscript.

- Ethics statement

- Data accessibility

<http://datadryad.org/submit?journalID=RSOS&manu=RSOS-192234.R2>

- Competing interests

- Authors' contributions

- Acknowledgements

- Funding statement

Because the schedule for publication is very tight, it is a condition of publication that you submit the revised version of your manuscript before 13-Jun-2020. Please note that the revision deadline will expire at 00.00am on this date. If you do not think you will be able to meet this date please let me know immediately.

Supplementary files will be published alongside the paper on the journal website and posted on the online figshare repository (<https://figshare.com>). The heading and legend provided for each supplementary file during the submission process will be used to create the figshare page, so please ensure these are accurate and informative so that your files can be found in searches. Files

on figshare will be made available approximately one week before the accompanying article so that the supplementary material can be attributed a unique DOI.

Finally, when resubmitting, please check the following email, which appears to be invalid:

ziyang.ling@student.manchester.ac.uk

Kind regards,
Lianne Parkhouse
Editorial Coordinator
Royal Society Open Science
openscience@royalsociety.org

on behalf of Dr Riccardo Avanzinelli (Associate Editor) and Andrew Dunn (Subject Editor)
openscience@royalsociety.org

Editorial Office requests:

As you have been requested to edit the written English, you must provide proof that you have done so: acceptable proof includes a certificate of language-editing from a language editing service or a signed letter from a native speaker of English. If you do not provide this proof, your manuscript may be returned to you.

For information about language editing services endorsed by the Royal Society, please follow the link below:

<https://royalsociety.org/journals/authors/language-polishing/>

Associate Editor Comments to Author (Dr Riccardo Avanzinelli):

The authors responded to and accounted for almost all the reviewers' comments and criticisms, and modified the manuscript accordingly. In my opinion the manuscript does not need further external reviews, but, from my own reading, some minor revision are still required before publications.

I am not an English mother tongue, but I found several typos and issues with the English language. For example, some of the modifications/additions made in response to the reviewers' comments, were done without ensuring that the sentences remained correct in English, resulting in broken or incomplete sentences. I listed, in the attached file, the issues I found throughout the ms (referring to the pages and lines), but I strongly recommend having the manuscript checked and proof-read by an English mother tongue or through professional English language editing.

Also, I am not convinced by the authors' response to one of the reviewers comments (Q9). The reviewer suggested a sentence to explain the use of the leachability index [$LX = -\log(Di)$], and the interpretation of its values. The authors added the LX parameter in the Table 7 (now Table 8), but did not modify the sentence at all. I strongly suggest to modify the sentence according to the reviewer suggestion, or at least to explain a bit better the use of the LX parameters, whose values are now provided in Tables 7 and 8, without too much discussion.

Overall my recommendation is Acceptance with Minor Revisions.

Author's Response to Decision Letter for (RSOS-192234.R2)

See Appendix E.

Decision letter (RSOS-192234.R3)

Dear Dr Shi,

It is a pleasure to accept your manuscript entitled "The rainfall effect onto solidification and stabilization of heavy metal-polluted sediments" in its current form for publication in Royal Society Open Science.

on behalf of Dr Riccardo Avanzinelli (Associate Editor)
openscience@royalsociety.org

Appendix A**ROYAL SOCIETY
OPEN SCIENCE****The rainfall effect onto solidification and stabilization of
heavy metal-polluted sediments**

Journal:	Royal Society Open Science
Manuscript ID	RSOS-190998
Article Type:	Research
Date Submitted by the Author:	05-Jun-2019
Complete List of Authors:	Sun, Yan; University of Shanghai for Science and Technology, Environmental Science and Engineering Zhang, Daofang; University of Shanghai for Science and Technology, Environmental Science and Engineering Li, Feipeng; University of Shanghai for Science and Technology, Environmental Science and Engineering Tao, Hong; University of Shanghai for Science and Technology, Environmental Science and Engineering Li, Moting; University of Shanghai for Science and Technology, Environmental Science and Engineering Mao, Lingchen; University of Shanghai for Science and Technology, Environmental Science and Engineering Gu, Zhujun; University of Shanghai for Science and Technology, Environmental Science and Engineering Ling, Ziyang; University of Manchester Institute of Science and Technology, School of Chemistry Shi, Huancong; University of Shanghai for Science and Technology, Environmental Science and Engineering
Subject:	environmental science < CROSS-DISCIPLINARY SCIENCES, Environmental engineering < ENGINEERING AND TECHNOLOGY, Geochemistry < EARTH SCIENCES
Keywords:	Leaching test, Solidification/stabilization, Diffusion coefficient, Heavy metals, Acid rain, Contaminated sediments
Subject Category:	Earth science

Author-supplied statements

Relevant information will appear here if provided.

Ethics

Does your article include research that required ethical approval or permits?:

Yes

Statement (if applicable):

We provided "ethics statement" in the text.

Data

It is a condition of publication that data, code and materials supporting your paper are made publicly available. Does your paper present new data?:

Yes

Statement (if applicable):

The original data was provided in the supplementary material. (ESM)

Conflict of interest

I/We declare we have no competing interests

Statement (if applicable):

CUST_STATE_CONFLICT :No data available.

Authors' contributions

This paper has multiple authors and our individual contributions were as below

Statement (if applicable):

The same information was provided in the text, author contribution.

- 1) Dr. Sun contributions to completed the main part of the experiments with data analysis and interpretation; the Rainfall test was introduced into the field, and the manuscript revision were completed. Dr. Sun also drafted the together with other corresponding authors Dr. Daofang Zhang, Dr. Feipeng Li and Dr. Huancong Shi.
- 2) Dr. Daofang Zhang agreement to be accountable for all aspects of the work in ensuring that questions related to the accuracy or integrity of any part of the work.
- 3) Feipeng Li provided the results of Figure 8-10, SEM and XRD with instruments. He provide the research grant of YRWEF201603.
- 4) Hong Tao helped to analysis the date of the semi-dynamic leaching tests , completed the linear regression of data Figure 5-7 .
- 5) Moting Li contributes to the preparation of specimens and analysis the date of the unconfined compressive strength (UCS) test and the toxicity characteristic leaching procedure (TCLP) tests, provided the results of Table 1-2, Figure 2 -4.
- 6) Lingchen Mao and Zhujun Gu provided background concept of Figure 1 and designed rainfall tests; they helped the development of the steps in Support information A.
- 7) Ziyang Ling helped Dr. Sun with the preparation of the sediment samples.

8) Dr. Huancong Shi completed several intensive revisions and submitted the manuscript.

i

The rainfall effect onto solidification and stabilization of heavy metal-polluted sediments

Yan Sun ^a, Daofang Zhang ^{a*}, Feipeng Li ^{a*}, Hong Tao ^a, Moting Li ^a, Lingchen Mao ^a,
Zhujun Gu ^a, Ziyang Ling ^b, Huancong Shi ^{a*}

- a. Institute of Environment and Architecture, University of Shanghai for Science and Technology, Shanghai 200093, China
- b. School of Chemistry, The University of Manchester, Oxford Rd, Manchester, UK. M13 9PL

Abstract:

Stabilized and solidified (S/S) with cement-based binder is an effective technique for treating heavy metal polluted soil and sediments. This paper presented an investigation of the effects of acid rain on leaching and hydraulic properties of cemented-based solidified contaminated sediments. The sediments from creeks in Chongming Island, Shanghai were selected as the contaminated artificial sediments and then subject to stabilization/solidification treatment using portland cement. Based on the effects of rainfall of acid rain test, the solidified sediments were cured after 3, 7, 14, and 28 days. The microstructural properties and heavy metal leaching behavior were also investigated. The experimental results indicated that the maximum unconfined compressive strength (UCS) of the sediments samples was 700 kPa. Compared with 760 kPa of standard curing conditions, the average loss of the unconfined compressive strength was 14.33% in the rainfall experiment. The toxicity characteristic leaching procedure (TCLP) tests indicated that the order of heavy metal stabilization rate was Pb > Cd > Cu. In the semi-dynamic leaching experiment, Cu still exhibited strong mobility after solidification, which was stronger at pH = 4 under acidic condition than at pH = 7 under neutral condition. After solidification and stabilization treatment, the heavy metal-polluted sediments were adequate to maintain the engineering strength and ensure environmental safety under the influence of rainfall.

Keywords: Sediments, Solidification and stabilization, Heavy metal, Rainfall

(Acid Rain)
Leachability
Microstructural properties - XRD, SEM
(Hydraulic properties)?
Mechanical properties - UCS

1. Introduction

The contamination of river sediments, especially heavy metal contamination, presents a serious threat to the environment with the rapid development of industry, in China. ⁽²⁾ Many rivers and lakes are polluted by the toxic metals such as lead (Pb), zinc (Zn), copper (Cu) and cadmium (Cd). If treated improperly, they can pose severe threats to the environment and human health. ^(3,4) Heavy metal are easy to accumulate but impossible to biodegrade. ⁽⁵⁾ Therefore, remediation of these contaminated sediments is of great concern for researchers.

There is an urgent need to develop cost-effective methods to remediate heavy metals contaminated sediments. One of the most efficient and low-risk remediation techniques is solidification/stabilization^{6,7}. S/S remediation and sediments stabilization is based on cement hydration reaction and commonly use cement-based binders^{8,9}. After S/S treatment, the heavy metals exist in the sediments matrix as metal hydrated phases, calcium-metal compounds and metal hydroxides, in a less soluble, mobile or toxic form by physical and chemical processes such as encapsulation, precipitation, adsorption and complexation. ^(10,11)

When the cement-based stabilized sediments matrix is exposed to rainfall or groundwater, leaching of heavy metals may occur. ⁽¹²⁾ Several studies have investigated this effect repeatedly. Okochi ~~H~~ et al.¹³ found that the anionic and cationic corrosive media such as H^+ , NH_4^+ , SO_4^{2-} , NO_3^- , Mg^{2+} in acid rain induced erosion damage to cement-based materials such as mortar and concrete. The ionic erosion media in acid rain diffuse into the cement-based materials, and react with solid components in the cement, so as to dissolve or decompose the hydrated products. Therefore, some soluble or insoluble salt minerals are produced, thus weakening the bonding ability and strength of the internal structure of the materials. Tang et al. ⁽¹⁴⁾ studied that H^+ could be neutralized with $Ca(OH)_2$ to reduce the alkalinity of materials, so that $C-S-H$ would be dissolved and corroded, and SO_4^{2-} reacts with hardened cement-based materials to produce expansive products such as $(CaSO_4 \cdot 2H_2O)$ and $CaAl_2Si_2O_8$. Kogbara ~~R~~ et al.¹⁵ found that when the material crystals were destroyed, heavy metals might leach out, and enhance the activity of Pb, Zn, Cd and other heavy metals. These heavy metal ions are released from solidified solids into interstitial water and water bodies, and do harm to the ecological environment. At the same time, other effects of rain water softening, scouring and leaching will also result in complicated erosion and destruction on cement-based materials, which cause potential hazards to the stability and ecology of slope and engineering matrix. Therefore,

REF?
ONLY Pb, Zn, Cu, Cd
STUDIED -
CONTAINED IN:
- CEMENT HYDRATE
- BINDER
- SEDIMENT

REF?

further research is needed on the safety and reliability of the technology.

The objective of this study was to investigate the effect of acid rain on hydraulic ^{MECHANICAL} ^{HYDRATED COMPOUNDS} ^{ELASTICITY?}
properties and leaching characteristics of cement-based solidified sediments. A series of
rainfall tests were performed on treated sediments samples to study the effects of rainfall
intensity on the leachability and microstructural properties of the treated sediments.
Simultaneously, the semi-dynamic leaching tests were performed on treated sediments
samples using simulated acid rain (SAR) as the extraction leachant with initial pH values of
4(0) and 7(0). The leaching data for selected heavy metals (Cu, Cd and Pb) were elaborated with
a diffusional leaching model based on penetration theory, which was aimed to identify the
release mechanisms of these heavy metals and to predict the long-term leaching behavior of
cement-based solidified sediments. Furthermore, X-ray diffraction (XRD) and scanning
electron microscope (SEM) analyses were performed to interpret the mechanisms controlling
the changes in these micro-properties.

2. Materials and Experimental procedures

The experiments consist of three parts: (1) the preparation of specimens, including
pretreatment and solidified samples; (2) field rainfall test and semi-dynamic leaching tests to
detect the behavior of heavy metals migration under acid rainfall conditions, (3) analysis of
the test data with XRD and SEM picture to evaluate the effect of rainfall on solidified
sediments, including mechanical properties, heavy metal leaching and the changes of micro-
structure.

2.1 Materials

The sediments used in this study were taken from Chongming District, Shanghai (E
121°17'38.6658", N 31°46'26.6154"). The properties of tested sediments were listed in Table 1.
The content of heavy metal was measured by inductively coupled plasma mass spectrometry ^{COMMENTS?}
(ICP-MS) (Perkin Elmer Nexion 300x) and the particle size distribution was measured by the ^{SAMPLING?}
intelligent laser particle size analyzer (BT-9300z). The cement was ^(WEIGHT, METHOD...) PO 42.5 ^{CEMI} Portland cement
(PC), purchased from Conch Cement (Shanghai) Co Ltd. The purity of the commercially
available chemical reagents was Analytical Grade of 99%, and they were glacial acetic acid,
nitric acid, perchloric acid, hydrofluoric acid and magnesium sulfate anhydrous.

Table 1 ~~The physical-mechanical~~ properties of tested sediment

Natural water content (%)	PH	Particle size distribution (µm)	Soil classification	+ voids
41.34 (PRECISION)	7.78 (PRECISION)	5-80	silty clay	+ ORGANIC MATTER CONTENT IF POSSIBLE

2.2 Preparation of specimens

2.2.1 Preparation of contaminated sediment

Cu, Cd and Pb were selected as the target heavy metals because they are commonly encountered toxic heavy metals found in sediments and soils worldwide. (6.1) To prepare contaminated sediments specimens, a predetermined volume of $Pb(NO_3)_2$, $Cu(NO_3)_2 \cdot 3H_2O$ and $Cd(NO_3)_2 \cdot 4H_2O$ solution with a known concentration of contaminant was added to the air-dried sediments until its water content reached about 45%. The (soil) and solution were thoroughly mixed for about 20 min using an electric mixer to create the homogeneous slurry. After maintenance in a closed container for 15 days at a temperature of $20 \pm 2^\circ C$, the sediments were evaporated at $60^\circ C$ under water bath, and the evaporation was not stopped until the water content reached 37-38%. The main content of sediment with the addition of heavy metals was presented in Table 2.

Table 2 ~~The total amount of heavy metals of tested sediment~~

Test indicators	Cd	Cr	Cu	Ni	Pb	Zn
INITIAL Content (mg/kg)	0.429	53.0	29.2	26.1	32.0	110.8
Content after pretreatment (mg/kg)	0.974	66.8	60.4	50.9	156.0	160.0

2.2.2 Preparation of solidified samples

The artificial contaminated sediment and cement were placed in a plastic bottle and were thoroughly mixed by an electric mixer and mixed at a speed of 120 r/min for 3 minutes firstly, then at a speed of 60 r/min for 1 minute to achieve homogeneity. The mixture was then poured into a cylindrical mold with diameter of 50 mm and height of 50 mm. After 24 hours' preservation, the sediments samples were carefully extruded from the mold using a hydraulic jack and subjected to standard curing condition (95% relative humidity and $22^\circ C$) for 7, 14, and 28 days.

2.3 Rainfall test

2.3.1 Field rainfall test

The field rainfall test was to place the standard cured samples in a beaker for different periods under natural rainfall conditions. From the beginning to the end of rainfall, the average period was 8 hours for the tests. The amount of rainfall was between 25-50 mm (heavy rain) or 50-100mm (rainstorm). Table 3 showed the actual pH and the amount of rainfall during the rainfall test. The detailed experimental procedures were provided in Support Information.

Table 3 The amount of rainfall and pH

The age of curing	3 days	7 days	14 days	28 days
The amount of rainfall (24 h/ml)	55.9	46.59	65.45	46.82
Rainfall level	heavy rain	heavy rain	heavy rain to rainstorm	heavy rain
pH	5.384	6.485	5.87	4.54

2.3.2 Semi-dynamic leaching test

Fig.1 is a schematic diagram of semi-dynamic leaching tests of stabilized sediments with cement solidification. The experimental procedures could refer to USEPA 1315 Method¹⁸ and ASTM 1308-08 Test¹⁹. Based on the average pH of 4.46 of rainfalls in Shanghai, two kinds of leachates were established with the aid of nitric acid solution (pH = 4) and deionized water (pH = 7), respectively. Under different initial pH values, the materials were divided into the benchmark group (non-renewal leaching group) and testing group (renewal leaching group). Four groups of parallel experiments were conducted for different pH values after 28 days of curing. In the leaching experiment, the volume of leachate was 1121 mL, the surface area of sample was about 118 cm², and the ratio of sample surface area to leachate volume was 1:9.5 (cm²·mL⁻¹). The simulated acid rain (SAR), used as the extraction leachate in the semi-dynamic leaching test, was prepared by diluting nitric acid (HNO₃) and ammonium sulfate ((NH₄)₂SO₄) in the deionized water. The stock solutions of SAR were adjusted to two pH value of 4.0 and 7.0 respectively.

Fig.1 Schematic diagram for semi-dynamic leaching tests

2.4 Test methods

The UCS tests as per ASTM D4219²⁰ with a fixed strain rate of 1%/min by means of a DYE-300S Model Compression and Fracture Resistance Integrative Machine (Huaxi Building Materials Experimental Instrument Co Ltd). A certain quantity of the fresh stabilized sediments was carefully sampled from the broken UCS specimen and then subjected to TCLP tests. Meanwhile, approximately 1 cm³ samples were freeze-dried then subjected to XRD and

SEM analyses. The Leachability of heavy metals was evaluated using the TCLP-EPA Method 1311. The initial pH of the TCLP extraction liquid (5.7 mL of CH₃COOH and 64.3 mL of 1 mol/L NaOH) was 4.93 ± 0.05.

3. Results and discussion

3.1 Effect of rainfall on mechanical properties of solidified and stabilized sediment

Rainfall experiments were performed on the specimens which were cured for 3, 7, 14 and 28 days, respectively, and Fig.2 demonstrated that the compressive strength of specimens after rainfall tests increased with the increased curing time, compared with the standard curing conditions as benchmark. The maximum compressive strength reached 760 KPa under standard curing conditions. After rainfall tests, the maximum compressive strength of the specimens was 700 KPa, with about 8% loss.

NUMBER OF SPECIMENS
UNCERTAINTIES
STANDARD DEVIATION

Fig.2 Unconfined compressive strength comparison of standard curing condition and rainfall conditions

To compare the compressive strength at different ages, the deterioration rate ϕ was a parameter of measurement, which is the ratio of compressive strength of specimens under standard curing conditions. The equation is 3-1:

$$\phi = \frac{f_0 - f_c}{f_0} \times 100\% \quad (3-1)$$

where ϕ is the deterioration rate of unconfined compressive strength (%), f_0 is the compressive strength of specimens under standard curing conditions (KPa), f_c is the compressive strength of specimens after rainfall (KPa). The comparison was performed at the same curing age.

Fig.3 showed the deterioration rate of the unconfined compressive strength of specimens after rainfall. With longer curing age, the loss of compressive strength was more reluctant in acid rain. The maximum deterioration rate was 34% of the compressive strength after 7 days of curing. The minimum rate was 2.89% after 28 days, and the average rate was 14.33%. When the specimen was eroded by acid rain, the hydrogen ion first reacted with the hydroxide colloid in the material, resulting in the dissolution of the gelation product and the decrease of the pH value. After 3 days of curing, the hydration reaction started to occur, a small quantity of

USH?
REF?

hydroxide colloids were dissolved and generated. The sulphate in rainwater infiltrated into the solidified material and reacted with $\text{Ca}(\text{OH})_2$ in cement to form gypsum, which filled the internal pores of the solidified material due to volumetric expansion, resulting in the decrease of porosity. Therefore, the ~~compressive strength~~ ^{UCS DEGRADATION RATE} of the specimens was 9.09% after 3 days of curing, ~~which was higher than that of the standard specimens.~~ ^{WITH UCS AFTER RAINFALL} After 7 days of curing, the hydration reaction was completed. A large number of products of $\text{Ca}(\text{OH})_2$ hydration were generated in the structure, and they were dissolved by H^+ under the action of rainfall. The material gradually lost alkalinity and the ~~compressive strength~~ ^{UCS / RESIDENCE?} dramatically decreased. After 28 days of curing, the skeleton was basically solidified and stabilized and the hydration products from cementing particles established a relatively stable structure. The hydration reaction of the tested block after 28 days of curing was relatively good. The decrease of the compressive strength was limited. Finally, rainfall was corrosive on the hydroxide colloids on the surface, but the corrosion had no significant effect on the ~~compressive strength~~ ^{(Q6)? REFERENCE? UCS}. The degree of damage was low after rainfall effect. Fig.3 indicated that the curing time of 28 days was the best among the rest.

Q: PORE? !?
REF?
Q: PORE INCREASE UCS? !?
NOT A UNDER ONLY DUE TO THE DECREASE OF PORE? !?
!?
!?
ALSO AT OTHER CONCENTRATIONS

Fig.3 Deterioration rate of ~~unconfined compressive strength~~ ^{UCS} after rainfall

3.2 Heavy metal leaching from solidified and stabilized sediment under rainfall

3.2.1 Leaching concentration of heavy metals

The stabilization effects of heavy metals were different before and after solidification and ~~stabilization~~ ^{AT 7 DAYS} because of different initial amount of Cd^{2+} , Cu^{2+} and Pb^{2+} . Therefore, the decreasing of leaching toxicity of heavy metals was determined by the stabilization rate. The definition of stabilization rate was equation 3-2.

$$\sigma = \left(1 - \frac{C_x}{C}\right) \times 100\% \quad (3-2)$$

where, σ is the stabilization rate of heavy metal (%), C_x is the total leaching amount of heavy metal X in solidified body ($\text{mg}\cdot\text{kg}^{-1}$), C is the total amount of heavy metal X in raw sediment ($\text{mg}\cdot\text{kg}^{-1}$).

NUMBER OF SPECIMENS?

Fig 4 (a-c) exhibited the total leaching amount of heavy metals in solidified/stabilized sediments and the stabilization rates under standard curing conditions and rainfall. The percentage on the column indicated the heavy metal stabilization rate. The order of heavy

metals' stabilization rates was $Pb > Cd > Cu$ of solidified body under standard curing
 conditions. The stabilization of Pb was the best, ^{BECAUSE} (for) it was not detected in the leaching test
 after solidification/and stabilization. The order of heavy metals' stabilization rates was still Pb
 $> Cd > Cu$ of solidified body after rainfall experiments. Pb was not detected in the leaching
 test after rainfall either. The stabilization rates of Cd and Cu were decreased by 2.3% and
 1.1%, respectively. ^{PRECISION?}

15 Fig.4 Variation of leached Cd^{2+} , Cu^{2+} and Pb^{2+} concentration with different curing
 16 conditions. ^{FROM THE 815 SEDIMENT} (The percentage on the column indicates the heavy metal stabilization rate)
 17 ^{AT 15 DAYS}

After solidification/and stabilization, the raw sludge contained lots of basic components. ^{TO DEVELOP OR}
 Heavy metal ions were converted from cations to insoluble precipitate, complex metal ^{DELETE}
 hydroxides, carbonate, residue and etc. The transformation of the morphology of heavy metals ^{REF?}
 also affected stabilization. The solidification/and stabilization effect of Pb was the best among
 these heavy metals. The morphology of Pb in crystals were relatively stable. It was found that
 Pb is the highest in the residual content of the ^{TO RATIO?} solidified stabilizer of 28 ^{DAYS} ¹⁵. The oxidized
 content of Fe and Mn decreases, and the carbonate and exchangeable state increases slightly²².
 Sari et al.²³ discovered that the adsorption equation of Pb is fitting with Freundlich equation
 and second order reaction kinetics of adsorption, indicating that the adsorption process of Pb
 is a typical ion exchange process. In the process of hydration, Pb^{2+} cations quickly permeate
 into the crystal structure of hydration product, to generate coralline crystal with $C_3S \cdot H_2O$.
 Despite the acid rain, there were little precipitation of metal ions after the corrosion of crystal.
 After ~~cement~~ solidification/and stabilization, Cd^{2+} was mainly in residual, followed by
 carbonate state and Fe-Mn oxidation state, the organic and exchangeable state were the least ^{REF?}
 content. After analyses, the carbonate and Fe-Mn oxidized states were more susceptible to acid ^{REF?}
 rain and easier to release heavy metals, which was consistent with the reduction of Cd^{2+}
 stabilization rate after rainfall tests. Under the strong basic environment, Cu increased rapidly
 in exchangeable and carbonate state. However, since Cu is a sulfophilic and organic matter
 element²⁵, the organic content accounts for the main part of the stabilized sediments. Some
 organic chemicals containing Cu are converted into residual state^{26,27} under basic conditions
 with high pH value.

The reasons for the low stabilization rate of Cu might be as follows: (1) The corrosion of ^{ADDITION}
 41 ^{IN CEMENT}

H⁺ and SO₄²⁻ onto carbonate and hydration products released Cu²⁺ cations into the water solution. (2) Within the strong base, Cu precipitated with the decomposition of organic compounds and was soluble in acidic solution. Meanwhile, Jiang²⁸ believed the competition did exist among several heavy metal ions as long as several heavy metal cations were mixed together. Generally, under the same conditions of adsorbent, time and initial concentration, the order of adsorption rate was Pb > Cd > Cu of the heavy metal ions. It was another proof for the lowest stabilization rate of Cu among these three heavy metal ions.

3.2.2 Semi-dynamic leaching experiment

The experiments indicated that the total amount of Cd²⁺ and Pb²⁺ was not detected as leached products from the surface of the sample, with cumulative leaching time. Therefore, the leaching of Cu²⁺ was discussed under simulated acid rain conditions.

(1) The dynamic leaching concentrations of Cu²⁺ with different leaching time

Fig. 5 demonstrated the concentrations of precipitated copper with cumulative leaching time. In general, at pH of 4 and 7, the concentration of Cu²⁺ increased with the increase of leaching time. When pH was 4, the leaching concentration of Cu²⁺ in the renewal group increased steadily in the first 96 hours, and then increased rapidly. This was possibly due to the existence of alkaline system in the solidified body. These alkalis neutralize some H⁺ in simulated acid rain solution. However, after the continuous corrosion of Ca(OH)₂ and hydration products by H⁺ in the cured body, the precipitation and adsorption of Cu²⁺ were gradually decreased in the cured body. The release and precipitation of Cu²⁺ resulted in the increase of concentration. When pH was 7, the variation trend of Cu²⁺ concentration was similar to that of pH = 4, and the maximum precipitation concentration was 15.86 μg/L. At pH of 4 and 7, the leaching concentration of Cu²⁺ increased rapidly in the control group during 0-24 hours. The rate was then slowly increased during 24-144 hours, and afterwards it increased rapidly again after 144 hours (6 days). During the first 24 hours, the concentration difference between the renewal group and the control group was very small. After 24 hours, the cumulative effect of the concentration in the control group turned to be manifest.

Fig.5 Variation of leached Cu concentration with leaching time

Fig.6 Variation of CFL(%) of Cu with leaching time

(2) The effective coefficient of diffusion

The diffusion coefficient of copper was calculated to analyze the dissolution characteristics of copper. The CFL formula for calculating cumulative-copper leaching rate under different leaching time is equation 3-3^{18,19}:

$$CFL = \frac{\sum m_n}{M_0} = \frac{\sum c_i \times V_{L,i}}{M_0} = \frac{M_t}{M_0} \quad (3-3)$$

where m_n is the pollutant leached from interval n during semi-dynamic leaching experiment (mg), c_i is the concentration of copper leached during the leaching (mg/L), $V_{L,i}$ is the volume of leachate, M_0 is the total amount of copper contained in raw sludge (mg), M_t is the cumulative mass of copper leached from solidified body surface after t time (mg).

The index D_e is another parameter to compare (material) mobility under different conditions. The definition is as follows^{18,19}:

$$D_e = \frac{\pi CFL}{4} \left[\frac{V}{S} \right] \quad (3-4)$$

where S is the surface area of solidified body (cm^2), V is the volume of solidified body (cm^3).

Fig 6 reflected the cumulative dissolution rate variation of Cu: CFL with $t^{1/2}$. Four curves of CFL- $t^{1/2}$ were divided into two groups according to the time nodes. From Fig.6, the CFL- $t^{1/2}$ linearity of pH = 4 and pH = 7 renewal group was divided into two sections at the critical time of 48 h, the correlation coefficients of the pH = 4 renewal group were $R^2 = 0.9831$ (PRECISION) during 0-48 h and $R^2 = 0.9909$ after 48 h, and the correlation coefficients of the pH = 7 renewal group were $R^2 = 0.9309$ during 0-48 h and $R^2 = 0.8845$ after 48 h. TO INDICATE ON GRAPH

Similarly, the CFL- $t^{1/2}$ linearity of pH = 4 and pH = 7 control group was divided into 24 h. The correlation coefficients of pH = 4 control group were $R^2 = 0.91$ during 0-24 h and $R^2 = 0.9872$ after 24 h, respectively. However, the correlation coefficient of pH = 7 control group was very low, namely $R^2 = 0.3936$. This curve was hard to be linear, for the Cu^{2+} was not detected before 144 h with only a small amount of leaching detected after 216 hours. In general, the fitting of simulated acid rain pH = 4 was better than that of pH = 7.

The slope was generated by the linear fitting of CFL- $t^{1/2}$. Under the same curing agent conditions, the initial pH value of simulated acid rain was the main factor that affected the slopes. From the graph, the slope of CFL- $t^{1/2}$ was more smooth with the increase of initial pH value. While for the renewal group under simulated acid rain conditions where pH = 7, the

slope increased with the increase of leaching time.

Table 6 Effect of pH Value of Leachate on Effective Diffusion Coefficient of Heavy Metal Cu²⁺

pH	Renewal group		Control group	
	t (h)	De (cm ² ·s ⁻¹)	t (h)	De (cm ² ·s ⁻¹)
4	0 ~ 216	8.86 × 10 ⁻¹⁰	0 ~ 24	8.41 × 10 ⁻¹¹
			24 ~ 216	1.82 × 10 ⁻¹⁰
7	0 ~ 144	7.22 × 10 ⁻¹²	0 ~ 24	2.27 × 10 ⁻¹³
	216	1.42 × 10 ⁻¹¹	24 ~ 216	1.20 × 10 ⁻¹⁰

Table 6 exhibited the effect of pH of leachate on effective diffusion coefficient of heavy metal Cu²⁺. The effective diffusivity (De) of copper in the renewal group was 8.86 × 10⁻¹⁰ cm²·s⁻¹ at pH = 4. The variation range of effective diffusivity (De) in the control group was 8.41 × 10⁻¹¹~1.82 × 10⁻¹⁰ cm²·s⁻¹. The effective diffusivity (De) of copper in the simulated rainfall pH = 7 renewal group was 7.22 × 10⁻¹² ~1.42 × 10⁻¹¹ cm²·s⁻¹, while that in the control group was 2.27 × 10⁻¹³~1.20 × 10⁻¹⁰ cm²·s⁻¹. Prica et al.²⁹ reported that the diffusion coefficients of heavy metals in solidified solids range from 10⁻¹⁰ cm²·s⁻¹ to 10⁻¹⁴ cm²·s⁻¹, of which the diffusion coefficient of copper is 10⁻¹³ cm²·s⁻¹, and the mobility of heavy metals is moderate.

Song et al.³⁰ reported that the diffusion coefficient of copper is 10⁻¹⁵ ~ 10⁻¹¹ cm²·s⁻¹. In the acidic condition, the corrosion of the solidified body was greater, so that the diffusion coefficient of pH = 4 was larger than that of pH = 7. Malviya et al.³¹ found that the mobility of heavy metals is high when De is smaller than 10⁻¹¹, the mobility of heavy metals is moderate when De is between 10⁻¹¹ and 10^{-12.5}, and the mobility of heavy metals is low when De is less than 10^{-12.5}. After comparison, in the renewal group at pH = 4, the migration of copper was larger, the diffusion coefficient De of copper was within the range of 10⁻¹⁰~10⁻¹¹, and the diffusion coefficient De of the control group increased with the time. At pH = 7, the migration of copper was moderate at 0-48 h, and increased quickly after 48 h. On the other hand, the migration of copper was small within 0-24 h. In the control group, the migration of copper increased with the accumulation of time. The migration of copper at pH = 4 (acidic) was higher than that at pH = 7 (neutral). The overall migration of copper was relatively large, indicating that the curing effect of copper was poor and the reason awaited further experimental

verification.

Fig.7 Leachate pH during the semi-dynamic leaching test (a) pH = 4, (b) pH = 7

(3) The regularity of pH value of leaching solution

Fig.7 showed the change of pH value of leaching solution vs leaching time. The pH
increased with the increase of leaching time from 0 to 216h. Within 0 to 24h, the pH of the
leaching solution increased significantly. Then within 24 to 120h, the pH value of the leaching
solution increased gradually and eventually reached an equilibrium after 120h. In the case of
pH = 4 and pH = 7 in the renewal group, the change of pH of the leaching solution was between
6.2 and 11.03. The pH value of the renewal group was higher than that of the control group in
general, because of the saturation of the alkaline medium released from the solidified body.
The change of pH value pH = 4 and pH = 7 demonstrated that the solidified body contained
acid-base buffer system, and the released OH⁻ neutralized H⁺, so that the final leaching solution
was basic. The variation of pH value was basically related to the release of Cu²⁺ from leaching
solution. REF?

3.3 Effect of rainfall on microstructures of solidified and stabilized sediment

Fig.8 the sediment XRD spectra before and after solidification

Fig. 8 showed the sediment XRD spectra before and after solidification. The main
components of the sediment were quartz and albite before solidification. After solidification,
certain hydration products were detected in the sediment, such as calcium carbonate, calcite
and hydrated calcium silicate.

Fig.9 SEM of test blocks for (a) standard curing and (b) 28^{DAYS} ~~28d~~ rainfall experiments

Fig 9 (a) demonstrated the ~~scanning electron microscopy (SEM)~~ of the specimens for 28
days under standard curing conditions. A large number of hydration products of acicular and
slab crystals were detected on the surface of the particles. Combined with XRD, these crystals
indicated that a large number of hydration products (ettringite and calcium hydroxide) were
broken down within the material structure. At the same time, the honeycomb like crystal
structure was observed on the surface of the particles of calcium silicate hydrate. After

1 comparison, Fig 9 (b) demonstrated the scanning electron microscopy (SEM) of the specimens
 for ~~28 days~~ ^{AFTER} under the rainfall tests. Some acicular crystals still existed, but the crystal structure
 of honeycomb was unstable and started to ~~demolish~~ ^{DESTROY}. Many flocculent crystal structures were
 observed as well. After rainfall, H^+ corroded some $Ca(OH)_2$, resulting in the decrease of crystal
 structure. The infiltration of SO_4^{2-} ~~demolished part of internal crystal composition~~. However,
 due to the hydration reaction which lasted 28 days, a relative stable skeleton was generated so
 that the crystals filled ^(-Pb) the voids. The structure had no significant impact on the ^{UCS} compressive
 strength of the test blocks, and the breakage rate was low. ^(SPECIMENS)

REF?
 REGENERATION
 BUT NOT DEGRADATION
 NEW TIMEPLANS

Fig.10 SEM of leaching filtrate pH = 4 and pH = 7 in semi-dynamic leaching experiment

Fig.10(a)(b) ~~were~~ ^{RESERVATION} the scanning electron microscopy (SEM) of leachate sample with pH
 = 4 and pH = 7. Fig.10(a) was the SEM of leachate sample with pH = 4, and the surfaces of
 the particles were covered with less flocculent crystals than that in Fig.10(b), and the crystal
 structure was visible. Compared with Fig 9(a) ~~as~~ the SEM under standard curing conditions,
 the crystal structure was significantly demolished. There were still flocculent or grid-like
 crystal structures attached to the surface of particles, but with few acicular crystals remained
 on the surface. The honeycomb crystal structure was reduced along with the agglomeration of
 large particles. The reduction of crystals in acidic environment was probably the main reason
 for ~~the~~ ^{RELEASE} ~~precipitation~~ in the leaching test. ~~Pb~~ and ~~Cd~~ were less affected by the reactions of
 hydration products, and their ~~fluidity~~ ^{RELEASE} was relatively low under the strong alkaline environment
 compared with acidic conditions. ^{→ EDD IEX?}

4. Conclusions

- SEDIMENT : SILTY CLAY ~~5180~~ 5180 μm
- CEMENT : CEM I 42.5
- CEMENT CONTENT
- INITIAL QUANTITIES OF Pb, Cd AND Cu

(1) For the rainfall experiment (daily rainfall is between 46-65 mm, and the average pH
 is 5.5), the corrosion of H^+ on hydroxide colloids and hydration products along with the
 infiltration of SO_4^{2-} ions resulted in the expansion erosion of large aggregates structure. The
 corrosion led to some losses of the ^{UCS} compressive strength of solidified body, and ~~the average~~ ^{NOT SIGNIFICANT}
 ~~loss was 14.33% of compressive strength~~ ^{UCS} and the maximum compressive strength was 700
 KPa. ^{→ DATA AT 7 AND 28 DAYS}
 51 ^{+ UCS AT 3 DAYS -}

(2) After the rainfall experiments, Pb^{2+} was not detected ~~at all~~. The order of heavy metal
 53 ^(UNDER DETECTION LIMIT OF ANALYSIS APPARATUS)

stabilization rates was $Pb > Cd > Cu$ of solidified solids after ^{27 DAYS OF} standard curing and ^{AFTER} rainfall
experiments.

(3) The Semi-dynamic leaching tests indicated that the migration rate of ^{Cu²⁺} copper at pH =
4 was higher than that at pH = 7, and the overall migration of ^{Cu²⁺} copper was relatively strong. ^(DATA)
Simple cement solidification method was not adequate to inhibit the migration of ^{Cu²⁺} copper,
which required further study. ^{ENVIRONMENTAL}
9 ^{THRESHOLDS?}

(4) Based on the analysis of SEM and XRD, the needle-like, slab-like and flocculent
14 ^(MINERALS) crystals were observed on the surface of the material particles after natural rainfall conditions
and semi-dynamic leaching experiments, indicating the existence of hydrated products inside
the structure, such as calcium hydroxide and calcium silicate hydrate. The material remained
intact structure and stability.

Funding Statement

This research was financially supported by the National Natural Science Foundation of
China (NSFC no. 41601229, 51679140, 21606150). This work was also supported by the key
laboratory of Yangtze River water environment, Ministry of Education of China
(YRWEF201603).

Data Accessibility

The datasets supporting this article have been uploaded as part of the Supplementary
Materials.

Competing Interests

We have no competing interests of this study.

Authors' Contributions

- 1) Dr. Sun contributions to completed the main part of the experiments with data analysis
and interpretation; the Rainfall test was introduced into the field, and the manuscript
revision were completed. Dr. Sun also drafted the together with other corresponding
authors Dr. Daofang Zhang, Dr. Feipeng Li and Dr. Huancong Shi.
2) Dr. Daofang Zhang agreement to be accountable for all aspects of the work in ensuring
that questions related to the accuracy or integrity of any part of the work.
3) Feipeng Li provided the results of Figure 8-10, SEM and XRD with instruments. He
provide the research grant of YRWEF201603.
4) Hong Tao helped to analysis the date of the semi-dynamic leaching tests , completed the

linear regression of data Figure 5-7 .

5) Moting Li contributes to the preparation of specimens and analysis the date of the
unconfined compressive strength (UCS) test and the toxicity characteristic leaching
procedure (TCLP) tests, provided the results of Table 1-2, Figure 2 -4.

6) Lingchen Mao and Zhujun Gu provided background concept of Figure 1 and designed
rainfall tests; they helped the development of the steps in Support information A.

7) Ziyang Ling helped Dr. Sun with the preparation of the sediment samples.

8) Dr. Huancong Shi completed several intensive revisions and submitted the manuscript.

Acknowledgements

We also thank Yongzheng Li for completion of original experiments but did not meet the
authorship criteria.

Ethics statement

We received ethical approval from a “lab safety and research ethics committee” of University
of Shanghai for Science and Technology to carry out our study. This university provide
consent in a Research Ethics and code of lab safety and ethics in scientific research and
publication.

References:

[1] Islam MS, Ahmed MK, Raknuzzaman M, Habibullah-Al-Mamun M, Islam MK. 2015 Heavy metal
pollution in surface water and sediment: A preliminary assessment of an urban river in a developing
country. *Ecol Indi.* **48**, 282-291. (doi:10.1016/j.ecolind.2014.08.016)
[2] TANDY S, BOSSART K, MUELLER R, et al. 2004 Extraction of heavy metals from soils using ~~to~~ homogenise.
biodegradable chelating agents. *Environ Sci & Technol.* **38**, 937-944. (doi:)
[3] Peng JF , Song YH , Yuan P , et al. 2009 The remediation of heavy metals contaminated sediment.
*Hazard Mater.* **161**, 633-640. (doi:)
[4] Kucuksezgin F, Uluturhan E, Batki H. 2008 Distribution of heavy metals in water, particulate matter
and sediments of Gediz River (Eastern Aegean). *Environ Monit & Assess.* **141**, 213-225.
(doi:10.1007/s10661-007-9889-6)
[5] Gopinath A, Nair SM, Kumar NC, Jayalakshmi KV, Pamalal D. 2010 A baseline study of trace metals
in a coral reef sedimentary environment, Lakshadweep Archipelago. *Environ Earth Sci.* **59**, 1245-
1266. (doi: 10.1007/s12665-009-0113-6)
[6] Alpaslan B , Yukselen MA . 2002 Remediation of Lead Contaminated Soils by
Stabilization/Solidification. *Water Air and Soil Pollution.* **133**, 253-263.
(doi: 10.1016/j.envint.2019.02.057)
[7] Fleri MA , Whetstone GT . 2007 In situ stabilisation/solidification: Project lifecycle. *Hazard Mater.*
**141**, 441-456. (doi:10.1016/j.jhazmat.2006.05.096)

[8] Singh TS, Pant KK. 2006 Solidification/stabilization of arsenic containing solid wastes using portland
cement, fly ash and polymeric materials. *Hazard Mater.* 131, 29–36.
(doi:10.1016/j.jhazmat.2005.06.046)
[9] Paria S, Yuet PK. 2006 Solidification--stabilization of organic and inorganic contaminants using
portland cement: a literature review. *Environ Rev.* 14. (doi:10.1139/A06-004)
[10] Guo B, Liu B, Yang J, et al. 2017 The mechanisms of heavy metal immobilization by cementitious
material treatments and thermal treatments: A review. *Environ Manage.* 193, 410.
(doi: 10.1016/j.jenvman.2017.02.026)
[11] Li XD , Poon CS , Sun H , et al. 2001 Heavy metal speciation and leaching behaviors in cement
based solidified/stabilized waste materials. *Hazard. Mater.* 82, 215-230.
[12] Voglar GE, Lestan D. 2010 Solidification/stabilization of metals contaminated in dust- trial soil from
former Zn smelter in Celje , Slovenia, using cement as a hydraulic binder. *Hazard.Mater.* 178, 926–
933. (doi:)
[13] Okochi H , Kameda H , Hasegawa SI , et al. 2000 Deterioration of concrete structures by acid
deposition - An assessment of the role of rainwater on deterioration by laboratory and field exposure
experiments using mortar specimens. *Atmos Environ.* 34, 2937-2945. (doi:10.1016/S1352-
2310(99)00523-3)
[14] TANG Xiyan, XIAO Jia, CHEN Feng. 2006 Effect and Research Progress of Acid Deposition on
Concrete Durability. *Mater Rev.* 20. (doi:)
[15] Kogbara RB , Al-Tabbaa A . 2011 Mechanical and leaching behaviour of slag-cement and lime-
activated slag stabilised/solidified contaminated soil. *Sci Total Environ.* 409, 2325-2335.
(doi: 10.1016/j.scitotenv.2011.02.037)
[16] Du YJ , Jiang NJ , Shen SL , et al. 2012 Experimental investigation of influence of acid rain on
leaching and hydraulic characteristics of cement-based solidified/stabilized lead contaminated clay.
*Hazard. Mater.* 225-226(none). (doi: 10.1016/j.jhazmat.2012.04.072)
[17] Hou D , He J , Lü, Changwei, et al. 2013 Distribution characteristics and potential ecological risk
assessment of heavy metals (Cu, Pb, Zn, Cd) in water and sediments from Lake Dalinouer, China.
*Ecotox Environ Safe.* 93, 135-144. (doi: 10.1016/j.ecoenv.2013.03.012)
[18]Epa U S.Method 1315 Mass transfer rates of constituents in monolithic or compacted granular
materials using a semi-dynamic tank leaching procedure.2009. (doi:)
[19] ASTM C1308-08 Standard method for accelerated leach test for diffusive release from solidified
waste and a computer program to model diffusive, fractional leaching from cylindrical waste forms.
2009. (doi:)
[20] ASTM standard D4219, Standard Test Method For Unconfined Compressive Strength Index Of
Chemical-Grouted Soils, American Society for Testing and Materials (ASTM), West Conshohocken,
PA, 2008. (doi:)
[21] USEPA Method 1311, Toxicity Characteristic Leaching Procedure (TCLP),United States
Environmental Protection Agency (USEPA), Washington, DC,1992. (doi:)
[22] Bao JP, Wang L, Xiao M. 2016 Changes in speciation and leaching behaviors of heavy metals in
dredged sediment solidified/stabilized with various materials. *Environ Sci Pollut R.* 23, 8294-8301.
(doi:10.1007/s11356-016-6184-5)
[23] Sari A, Tuzen M, Citak D, Soylak M. Equilibrium.2007 Kinetic and thermodynamic studies of
adsorption of Pb(II) from aqueous solution onto Turkish kaolinite clay. *Hazard. Mater.* 149, 283-291.
(doi:10.1016/j.jhazmat.2007.03.078)

- [24]Xue Q , Wang P , Li JS , et al. 2017 Investigation of the leaching behavior of lead in
stabilized/solidified waste using a two-year semi-dynamic leaching test. *Chemosphere*, **166**, 1-7.
(doi:10.1016/j.chemosphere.2016.09.059)
[25] Saussaye L, Hamdoun H, Leleyter L, van Veen E, Coggan J, Rollinson G, et al. 2016 Trace element
mobility in a polluted marine sediment after stabilisation with hydraulic binders. *Mar Pollut Bull.* **110**,
401-408. (doi:10.1016/j.marpolbul.2016.06.035)
[26] Dermatas D, Meng X. 2003 Utilization of fly ash for stabilization/solidification of heavy metal
contaminated soils. *Eng Geol.* **70**, 377-394. (doi:
[27] Shon CS, Estakhri CK, Lee D, Zhang D. 2016 Evaluating feasibility of modified drilling waste
materials in flexible base course construction. *Constr Build Mater.* **116**, 79-86.
(doi: 10.1016/j.conbuildmat.2016.04.100)
[28] Jiang MQ, Jin XY, Lu XQ, Chen ZL. 2010 Adsorption of Pb(II), Cd(II), Ni(II) and Cu(II) onto
natural kaolinite clay. *Desalination.* **252**, 33-39. (doi:10.1016/j.desal.2009.11.005)
[29] Prica M, Dalmacija B, Dalmacija M, Agbaba J, Krcmar D, Trickovic J, et al. 2010 Changes in metal
availability during sediment oxidation and the correlation with the immobilization potential. *Ecotox*
*Environ Safe.* **73**, 1370-1377. (doi:10.1016/j.ecoenv.2010.06.014)
[30] Song F, Gu L, Zhu N, Yuan H. 2013 Leaching behavior of heavy metals from sewage sludge
solidified by cement-based binders. *Chemosphere.* **92**, 344-350.
(doi: 10.1016/j.chemosphere.2013.01.022)
[31] Malviya R, Chaudhary R. 2006 Leaching behavior and immobilization of heavy metals in
solidified/stabilized products. *Hazard. Mater.* **137**, 207-217. (doi: 10.1016/j.jhazmat.2006.01.056)

Fig.1 Schematic diagram for semi-dynamic leaching tests

Fig.2 ^{UCS} Unconfined compressive strength comparison of standard curing condition and ~~AFTER rainfall conditions TEST~~

→ UNCERTAINTIES?
STANDARD DEVIATION?
REPEATABILITY?

Fig.3 Deterioration rate of ^{UCS}unconfined compressive strength after rainfall

Fig.4 Variation of leached Cd²⁺, Cu²⁺ and Pb²⁺ concentration with different curing conditions. The percentage on the column indicates the heavy metal stabilization rate.
 FROM THIS IS DERIVED
 AT 14 DAYS

Fig.5 Variation of leached Cu concentration with leaching time

Fig.6 Variation of CFL (%) of Cu with leaching time

Fig.7 Leachate pH during the semi-dynamic leaching test

(a) (b)

Fig.8 the sediment XRD spectra before and after solidification

Fig.9 SEM of test blocks for (a) standard curing and (b) ~~28d~~ ^{AFTER} rainfall experiments ~~28d~~ ^{3 DAYS}

Appendix B

Dear Dr Shi:

Manuscript ID RSOS-190998 entitled "The rainfall effect onto solidification and stabilization of heavy metal-polluted sediments" which you submitted to Royal Society Open Science, has been reviewed. The comments from reviewers are included at the bottom of this letter.

In view of the criticisms of the reviewers, the manuscript has been rejected in its current form. However, a new manuscript may be submitted which takes into consideration these comments.

Please note that resubmitting your manuscript does not guarantee eventual acceptance, and that your resubmission will be subject to peer review before a decision is made.

Your resubmitted manuscript should be submitted by 01-Apr-2020. If you are unable to submit by this date please contact the Editorial Office.

Kind regards,
Anita Kristiansen
Royal Society Open Science Editorial Office
Royal Society Open Science
openscience@royalsociety.org

on behalf of Dr Riccardo Avanzinelli (Associate Editor) and Jon Blundy (Subject Editor)
openscience@royalsociety.org

Associate Editor Comments to Author (Dr Riccardo Avanzinelli):

Associate Editor:

Comments to the Author:

Dear Dr Huancong,

Your manuscript has been examined by two experts who provided important criticisms and overall negative reviews, recommending major revision and outright rejection, respectively.

Reviewer #1 found the experimental approach interesting, but found several major issues including i) the lack of several important details and information on the experiments that were not provided in the manuscript, and ii) the insufficient discussion and explanation of some important aspects of the experiments and their implications.

Reviewer #2 rejected the paper stating that it failed to reference several recent studies made on the same subjects and stating the discussion section is made without solid evidence and scientific explanation.

From my own reading of the paper, I agree with the reviewers criticisms, and I also found other major issues, especially

regarding the description of the analytical methods used in this study. The manuscript focuses on the mobility of toxic elements (Cu, Pb, Cd) in sediments after stabilization with cement-based binder and most of the discussion is based on measurements of the concentration of these elements in the samples before and after the experiments. However no indications are provided on the procedures or even the analytical methods used to determine such concentrations, and the resulting analytical errors.

Therefore, according to these reviews and from my own evaluation, I believe the manuscript has too many shortcomings and it does not meet the standard for publication on RSOS.

My recommendation is **“Reject and allow resubmission”**, considering that the experimental approach is interesting (as stated also by reviewer #1) and that the study has the potential to provide a useful contribution, but only if the manuscript is completely restructured and significantly implemented taking into consideration the reviewers' comments and criticism.

Response:

The editor and both reviewers have brought many useful suggestions and comments, we were very grateful, and we have conducted substantial modifications and corrections onto this manuscript and some of the experiments were re-designed and some experimental results were replaced with new completed data and information.

The main changes were outlined as follow:

- 1. To editor, almost 60% of the manuscript was corrected with new references and novel ideas provided into the manuscript in Introduction, Abstract, and Conclusion.**
- 2. To reviewer 1, the Experiments of filed rainfall test was improved and missing information was provided.**
- 3. To reviewer 2, the Experiments of semi-dynamic leaching test was modified and re-designed. The new experiments were provided to replace the old one with new results that Fig 5 were supplemented and other charts have been refined. The evidence was strengthened and the scientific interpretation was provide onto the products.**
- 4. Afterward, based on the revised manuscript, the novelty of this study was summarized as follows:**
 - The effect of rainfall on the curing process of solidified contaminated sediment was investigated by filed rainfall test.**
 - The leaching characteristics of heavy metals in solidified sediment were investigated by semi-dynamic leaching test, its diffusion coefficient was calculated, and the long-term safety of solidified sediment was evaluated.**
 - The leaching mechanism of each leaching period was identified first, and the diffusion coefficient of each leaching period was calculated which was diffusion control, the average diffusion coefficient was calculated to ensure the accuracy.**

After these changes, the quality of the manuscript was improved according to the reviewer's suggestion and comments. We have try our best to edit and improve this work, to the best of my knowledge. We are very grateful for the opportunity of revision.

Reviewers' Comments to Author:

Reviewer: 1

Comments to the Author(s)

The manuscript presents the study of a river sediment, artificially contaminated, the treated with Portland cement, and the potential of stabilization of Cu, Cd and Pb.

1, The experimental approach is interesting but there are some missing information to help at the complete comprehension of the involved stabilization processes.

Response:

The experiments of filed rainfall test were modified and completed. The missing information was added and supplemented into the results to help to complete the comprehension of the stabilization process.

2, All the comments are noted on the manuscript, but some comments are given below. It is necessary to precise :the physical parameters of the river sediment. PSD: no particles under 5 µm? no clayey particles (0/2 µm)? So, the contamination is not associated to clay fraction

Response:

According to particle size, the samples contain only sand and powder, but little clay minerals and secondary minerals. The experimental sediment consists of 3.7% clay particles, 70.9% silt particles, 25.4% Sand particles.

3, Atterberg limits : liquidity limit and index plasticity are important to define the clay activity of the sediment, defined as a silty clay

Response:

As clay particles account for 0~15%, silt particles account for 45~100%, and sand particles account for 0~55%, experimental sediment is defined as silt loam. Physical properties such as liquid limit and plastic limit were shown in **table 1**.

Changes: 2nd paragraph Section 2.1

Table 1 The physical-mechanical properties of tested sediment

Natural water content (%)	pH	Particle size distribution (µm)	organic content(%)	Liquid limit (%)	Plastic limit (%)	Plasticity Index
41.34	7.78	5-80	2.2	33.9	22.6	11.3

4, Organic matter content if possible, the organic fraction being defined as important in the stabilization processes

Response:

In this study, the organic matter content of sediment was not high (2.2%, organic matter is determined by loss on ignition), which was not considered to affect the stabilization processes significantly in this research.

5, The sampling method of the river sediment (weight, mode...)

Response:

According to 'Water quality - Sampling - Part 12: Guidance on sampling of bottom sediments(ISO 5667-12-1995)', a columnar sediment sampler was used to collect samples, after removed the lower and upper sediment, the sediment in the middle of the sampling column (about 20 cm) was stored in a special refrigerator, which was taken back to the laboratory for testing and analysis.

[1] ISO 5667-12-1995. Water quality - Sampling - Part 12: Guidance on sampling of bottom sediments [S].

6, As soon as possible, that the contamination of the sediment is artificial: how the addition concentrations are defined?

what it means (contamination, pollution, waste, reuse...)?

Response:

The addition concentration is based on China's specification "GB 15618-2018. Soil environment quality risk control standard for soil contamination of agriculture land".

[2]GB 15618-2018. Soil environment quality risk control standard for soil contamination of agriculture land [S]

7. Only Cu, Cd and Pb are studied: why not Cr, Ni and Zn, which are also analyzed ?

Response:

Considering that there are many heavy metals that are seriously polluted and worth of our study, but we cannot study all heavy metals in practice. So we selected three typical and representative heavy metals, Pb, Cu and Cd.

In this study, Pb was chosen as the target heavy metal contaminant because it is one of the most toxic elements to human health, and a growing number of sites have been contaminated by Pb in China [1]. Within contaminated sediment, Pb speciation is controlled by: (1) specific adsorption or ion exchange adsorption to soil mineral matrix; (2) precipitation of sparingly soluble compounds; (3) formation of complexes with organic matter [2].

Cu²⁺ has been widely used for petroleum, paper, and paperboard processing, ceramics manufacturing, electroplating, glass manufacturing, fertilizer production, etc., which also induced severe environmental pollution such as soot emitted from copper-zinc ore smelting and wastewater discharged from electroplating and metal processing [3], [4]. For human beings, higher concentrations (>5 mg/L) of Cu in the body has been associated with severe renal failure and liver diseases [5].

Cadmium (Cd) is among the most toxic heavy metals in the world and is acknowledged as a non-essential element to plants and animals [6]. Compared to other heavy metals, Cd is more readily accumulated in rice plants [7]. In China, a large area of Cd-contaminated paddy soil is still used to cultivate rice because of limited land resources and the enormous food demand from a large population [8]. Cd accumulation in rice grains poses an enormous potential health risk to people through the food chain, even to those who are not living in areas under high soil Cd pollution [9].

While the total amount of heavy metals detected was lower than the Risk screening value (according to "soil environmental quality of agricultural land soil pollution risk control standards"), to more clearly understand the effect of cement stabilization of heavy metals in sediment solidifying Cu, Cd and Pb are chosen as the additional metal.

[1] Y.S. Cui, X. Du, Li.P. Weng, W.H. Van Riemsdijk, Assessment of in situ immobilization of lead (Pb) and arsenic (As) in contaminated soils with phosphate and iron: solubility and bioaccessibility, *Water Air Soil Pollut.* 213 (2010) 95–104

[2] P.E. Jensen, L.M. Ottosen, A.J. Pedersen, Speciation of Pb in industrially polluted soils, *Water Air Soil Pollut.* 170 (2006) 359–382.

[3] J.G. Dean, F.L. Bosqui, K.H. Lanouette, Removing heavy metals from waste water *Environ. Sci. Technol.*, 6 (6) (1972), pp. 518-522.

[4] M.R. Gandhi, G.N. Kousalya, S. Meenakshi, Removal of copper(II) using chitin/chitosan nano-hydroxyapatite composite *Int. J. Biol. Macromol.*, 48 (1) (2011), pp. 119-124.

[5] R.R. Pawar, Lalmunsiam, H.C. Bajaj, S.-M. Lee, Activated bentonite as a low-cost adsorbent for the removal of Cu(II) and Pb(II) from aqueous solutions: batch and column studies *J. Ind. Eng. Chem.*, 34 (2016), pp. 213-223.

[6] J. de Livera, M.J. McLaughlin, G.M. Hettiarachchi, Cadmium solubility in paddy soils: effects of soil oxidation, metal sulfides and competitive ions *Sci. Total Environ.*, 409 (2011), pp. 1489-1497

[7] S. Uruguchi, S. Mori, M. Kuramata, A. Kawasaki, T. Arao, S. Ishikawa, Root-to-shoot Cd translocation via the xylem is the major process determining shoot and grain cadmium accumulation in rice, *J. Exp. Bot.*, 60 (2009), pp. 2677-2688

[8] L.L. Yu, J.Y. Zhu, Q.Q. Huang, D.H. Su, R.F. Jiang, H.F. Li, Application of a rotation system to oilseed rape and rice fields in Cd-contaminated agricultural land to ensure food safety, *Ecotoxicol. Environ. Saf.*, 108 (2014), pp. 287-293

[9] M.S. Rodda, G. Li, R.J. Reid, The timing of grain Cd accumulation in rice plants: the relative importance of remobilisation within the plant and root Cd uptake post-flowering, *Plant Soil*, 347 (2011), pp. 105-114

8, The Portland cement 42.5 content used (by weight of dry sediment)

Response:

The quality ratio of cement to the dry contaminated sediment was 1:4.

9, The objective of the solidification: UCS28 = 0.76 MPa is weak for subgrade layer for example

Response:

According to the Chinese design code “regulations in the design code for highway subgrade (JTGD30-2015)”, the UCS index of the lightweight soil of the subgrade should be greater than 600 kPa for expressways and first-class highways, and more than 500 kPa for secondary and sub-secondary highways. The UCS of the solidified sediment meets the specification requirements, so the heavy metal contaminated sediment can be used as roadbed filler after solidified and stabilized.

Koenig A^[1] suggest that the UCS of waste should be greater than 50 kPa for landfill treatment, while the resource conservation and recovery act (RCRA) of the United States suggests that it should be greater than 300 kPa^[2]. In this study, the recommended value of RCRA is 300 kPa, so the curing energy can meet the requirement of building backfill strength.

[1] Koenig A, Kay J N, Wan I M. Physical properties of dewater sludge for landfilling [J]. *Water Science and Technology*, 1996, 34:533-540.

[2] Shi C, Fernández-Jiménez A. Stabilization/solidification of hazardous and radioactive wastes with alkali-activated cements [J]. *Journal of hazardous materials*, 2006, 137(3): 1656-1663.

10, The Portland cement content in Cu, Cd and Pb: Cu in an usual element of the cements and it is important to determine is addition

Response:

The cement contents of Cu, Cd and Pb are 24.3, 0.135, 26.5 mg/kg which were shown in end of **table 2**

Changes: 2th paragraph Section 2.1

Table 2 The total amount of heavy metals of tested sediment and cement

Test indicators	Cd	Cr	Cu	Ni	Pb	Zn
Sediment content (mg/kg)	0.429	53.0	29.2	26.1	32.0	111
Sediment content after pretreatment (mg/kg)	0.974	66.8	60.4	50.9	156	160
Cement content(mg/kg)	0.135	19.6	24.3	12.1	26.5	30.2

11, The number of samples/specimens by test: then the figures have to present the standard deviations or uncertainties of the measurements

Response:

We prepared three groups of parallel samples and took the average values. The error bars were shown in the chart of Figure 2 and 5.

Changes: 1st paragraph Section 3.1 and 4th paragraph Section 3.2.3

Fig.2 UCS comparison of standard curing condition and rainfall conditions

Fig.5 CFL of target heavy metals (%) under different pH conditions: (a) Cu; (b) Pb; (c) Cd.

12, as soon as possible that the mechanical performance measured is unconfined compressive strength (UCS) (not hydraulic properties)

Response:

UCS is an important index to measure the curing effect and evaluate the engineering properties of the cured samples. It is included into the text.

13, The dimensions of the samples on which UCS tests are realized: it is recommended to realize UCS test on samples of slenderness 2 (high/diameter)

Response:

The stress-strain curves were calculated when the ratio of height to diameter was 0.5, 1.0, 1.5, 2.0 and 2.5, while keeping other parameters unchanged, and the number of cell height directions varied with the height to ensure the cell size remained unchanged. Taking the strength q_{u0} at the ratio of height to diameter of 2 as the reference value, through regression analysis, the ratio of q_u to q_{u0} at any ratio of height to diameter of H/D is obtained as follows:

$$q_u/q_{u0} = 0.983(H/2D)^{-0.455}$$

-14, The references for some affirmation (noted “REF?”) or the standards used for samples confection, conservation, the SAR, etc.

Response:

Based on the average pH of 4.46 of rainfalls in Shanghai, the simulated acid rain (SAR), used as the extraction leachate in the semi-dynamic leaching test, was prepared by diluting nitric acid (HNO_3) and ammonium sulfate ($(NH_4)_2SO_4$) in the deionized water. The stock solutions of SAR were adjusted to two pH value of 2.0, and 4.0 respectively.

[1] Du YJ, Wei M L, Reddy K R, et al. Effect of acid rain pH on leaching behavior of cement stabilized lead-contaminated soil[J]. J Hazard Mater, 2014,271:131-140.

15, in conclusion, first, the materials used for the experimental study, before the synthesis of the results Moreover it is necessary to explain/discuss of: The impact of a decrease of 8% of the UCS after acid rainfall: what are the technical applications?

Response:

According to the recommendations of Cembureau (European cement statistics and technical association), the chemical erosion of concrete by water and soil containing corrosive substances is evaluated as weak erosion with a pH of 5.5-6.5. Moderate erosion occurs at a pH of 4.5-5.5. Strong erosion occurs when the pH value is 4.0-4.5. If the pH value is less than 4.0, it is extremely strong erosion.

These acidic precipitation settle to the surface of the building after slow deposition, when there is water in the presence of the situation (such as rain, fog, dew), will produce acidic erosion of the building, the period of compressive strength decline, the project should try to avoid acid rain corrosion.

According to ACI building code 318-3, exposure to sulfates can be divided into four different hazard levels, and the protection requirements are listed:

Negligible erosion: when the sulfate content in the particles is less than 0.1% or the water is less than 150ppm (mg/l), there is no limit on the type of cement to the water-cement ratio;

Moderate erosion: when the sulfate content of the particles is 0.1 ~ 0.2% or 1500 PPM in water, pozzolan Portland cement or slag cement should be applied, and the water-cement ratio of ordinary concrete should be less than 0.5;

Severe erosion: when the sulfate content in the particles is 0.2 ~ 2.0% or 1500 ~ 10000ppm in the water, sulfate-resistant cement should be selected with the water-cement ratio less than 0.45.

Very serious erosion: when the sulfate content in the particles is more than 2.0% or more than 10000ppm of water, should choose sulfate resistant cement admixture with volcanic ash admixture, with less than 0.45 water-cement ratio.

[1] Xie SD, Zhou D, Yue QX, Liu HL. Effect of simulated acid rain on concrete [J]. Environmental Science, 1995(05):22-26+92.

16, The impact of a weak stabilization: what are the technical (chemical) referential? Environmental thresholds? what is the interest of the diffusion equations with the technical applications?

Response:

Malviya and Chaudhary pointed out that when the observation diffusion coefficient $D_i^{obs} < 3 \times 10^{-13} \text{ m}^2/\text{s}$, the pollutant migration is low. When $3 \times 10^{-13} \text{ m}^2/\text{s} < D_i^{obs} < 10^{-11} \text{ m}^2/\text{s}$, the pollutant migration is moderate. When $D_i^{obs} > 10^{-11} \text{ m}^2/\text{s}$, pollutant migration is high.

In order to interpret the results, the negative log of observation diffusion coefficient ($-\lg D_i^{obs}$) can be taken to define the Leachability index of heavy metals after correction. Environment Canada has listed this parameter as a control index for the utilization and disposal of solidified and stabilized contaminated soils. The formula for LX is shown below. When LX is greater than 9, it indicates that the curing and stabilization treatment of contaminated soil is effective and can be used for controlling the curing stop, such as roadbed filler. When LX is greater than 8, the solidified soil can be disposed of by landfill. When LX is less than 8, the solidified soil is not suitable for landfill disposal.

$$LX = \frac{1}{n} \sum_{i=1}^n (-\lg D_i^{obs})$$

[1] Malviya R, Chaudhary R. Evaluation of leaching characteristics and environmental compatibility of solidified/stabilized industrial waste [J]. Journal of Material Cycles and Waste Management. 2006, 8(1):78-87.

[2] Moon D H, Dermatas D. An evaluation of lead leachability from stabilized/solidified soil under modified semi-dynamic leaching conditions [J]. Engineering Geology. 2006, 85(1-2):67-74.

17, Are specific forms of erosion observed on the samples/specimens surface? does the leaching reach the

sample/specimen core (sufficient porosity or permeability)?

Response:

All of the previous studies suggest that diffusion is the controlling mechanism of most heavy metals that are leaching from S/S treated sediments including this research [1], which is not involved in the specimen core.

[1] Yan-Jun Du, Ming-Li Wei, Krishna R. Reddy, Zhao-Peng Liu, Fei Jin. Effect of acid rain pH on leaching behavior of cement stabilized lead-contaminated soil[J]. Journal of Hazardous Materials, 2014, 271.

18, With the obtained results, the intact (micro)structure and the stability of the cement treated sediment

Response:

We focused on the mechanical properties and leaching toxicity of the samples instead of intact (micro) structure.

19, The use of “stabilization” term to S/S sediment supposes that the stabilization is effective. The gypsum $\text{CaSO}_4 \cdot 2\text{H}_2\text{O}$ is not an expansive mineral; ettringite is. If gypsum is formed after rainfall on the treated sediment after 3 days of cure, it could decrease the porosity. Is it observed by DRX or SEM/EDS-EDX? The gypsum is not an hydrate which can bind the soil particles. The processes of dissolution of the minerals lead to recombination in new minerals but not in total destruction.

Response:

The sulfate in rainwater (SO_4^{2-}) reacted with $\text{Ca}(\text{OH})_2$ in cement so as to generate gypsum which then reacted with hydrated calcium aluminate to generate ettringite. The volume of ettringite was magnified several times than initial hydration product, which increased the inner stress of the solid phases. When the erosion proceeded, the specimen would release the inner substances through cracks on its surface. Therefore, the compressive strength of the specimens was 9.09% after 3 days of curing, which was higher than that of the standard specimens.

20, Some references to analyse the results are not adapted to the context. For example, it is probably not relevant to compare the geochemical behavior of a kaolinite clay to that of a silty clay.

Response:

Relevant references have been removed, and new references have been inserted.

Changes:

Ref 13, 14, 15, 35 were introduced into the text.

21, On the form:

-use the present form if possible, rather than the past form

-take care of the precision of the data

- take care of inconsistencies between the abstract and the manuscript or the submission: keywords of the presentation page are different of keywords of the abstract; samples/specimens subjected to 3 days of cure; presentation of Zn as toxic metal (metalloid) in introduction, not studied after...

- replace:

o “KPa” by “kPa”

o “PH” by “pH”

o “crystals” by “minerals”

- see manuscript annotations

Response:

Relevant errors have been corrected

Reviewer: 2

Comments to the Author(s)

Review of RSOS-190998 "The rainfall effect onto solidification and stabilization of heavy metal-polluted sediments"

General comments:

This study aims to investigate the effect of acid rain on hydraulic properties and leaching characteristics of cement-based solidified sediments.

The introduction of the manuscript should be reshaped. And it is important to introduce how the research is built on relevant studies and what is the research objective.

Response: The reviewer's suggestion is well taken, and we had completed substantial modification onto the manuscript. There are 50% of the contents changes, and the summary was described here:

1. The Experiments of semi-dynamic leaching test was re-designed and new experiments were provided.
2. Fig 5 were supplemented. The Table 6 were replace with Table 5, Table 4,6 were supplemented, and other charts have been refined.
3. The detailed Design of experiments included these factors
 - a) The Exp 2.3 was replaced with 2.2.2 Semi-dynamic leaching test
 - b) The research objective was changed to investigate the effect of rainfall on mechanical properties and leaching characteristics of cement-based solidified sediments because before the solidified contaminated sediment is recycled in the environment, it is necessary to investigate the effect of complex conditions on leaching behavior of them, with operation conditions
 - c) The results of the new experiments were discussed into Section 3.2, as solid evidence. Furthermore, the scientific explanation were also provided as follows.

As to my knowledge, the effect of acid rain on cement-based solidified sediments is widely documented in literatures. The novelty and academic merits of this study are not properly justified. The structures and contents of the manuscript should be further improved. In addition, the present form does not have sufficient results presenting the novelty of a high-quality journal paper.

Response:

Novelty:

- ◆ The effect of rainfall on the solidification process of heavy metals contaminated sediment by cement was investigated through field rainfall test.
- ◆ The leachability of heavy metals in solidified sediment were investigated by semi-dynamic leaching test, its diffusion coefficient was calculated, and the long-term safety of solidified sediment was evaluated.
- ◆ The leaching mechanism of each leaching period was identified first, and the diffusion coefficient of each leaching period of diffusion control was calculated, the average diffusion coefficient was calculated to ensure the accuracy.
- ◆ To calculate the diffusion coefficient, the leaching mechanism of each leaching period was identified first. Then leaching periods of diffusion control were selected to calculate their diffusion coefficients and the average diffusion coefficient was calculated to ensure the accuracy.

I believe the manuscript should be rejected. Specific comments are provided as follows:

Specific comments:

(1) Introduction:

Many references cited in the manuscript are more than 10 years ago. Thus, it does not properly represent advances in scientific knowledge. Recent papers about sediment in the construction industry in 2015-2019 should be critically reviewed (e.g., Cement Concr. Compos. 2019, 104, 103348; J. Clean. Prod., 2018, 199, 69-76) in the manuscript.

Response:

The introduction has been rewritten and the latest references cited. The experiments have been redesigned and conducted according to the experimental progress.

(2) Materials & methods:

Page 4: Please provide the cement ratio in the solidified samples. This is important to the performance of the solidified samples.

Did the authors consider the influence of organic matter in the sediment on the cement hydration?

Response:

The quality ratio of cement to the dry contaminated sediment was 1:4. In this study, the organic matter content of sediment was not high (2.2%, organic matter is determined by loss on ignition), mainly humus formed from animal and plant residues and animal excreta, and a small amount of low-molecular organic acid, which was easily biodegradable and not considered in this research.

(3) Discussion:

Page 12-13: There are so many paragraphs in this section just simply discuss the experimental results without solid evidence and scientific explanation.

Tables, figures should be improved. Key results are not well presented and it is very hard to follow.

Response:

We focused on the mechanical properties and leaching toxicity of the samples instead of intact (micro)structure. New experimental data have been analyzed.

Changes:

Updated Abstract:

Rainfall makes impacts on process of solidification/stabilization as well as the long-term safety of solidified matrix. In this study, the effect of rainfall on solidification/stabilization process was investigated by rainfall test. The unconfined compressive strength (UCS) and toxicity characteristic leaching procedure (TCLP) were adopted to characterize the properties of S/S sediments before and after rainfall test. The samples cured for 28 days were subjected to semi-dynamic leaching tests with a simulated acidic leachant prepared at pH of 2.0, 4.0 and 7.0. Effectiveness of S/S treatment was evaluated by diffusion coefficient (D_e) and leachability index (LX). The results indicated that UCS decreased by 14.33% as average affected by rainfall, while the rainfall had little effect on the leaching characteristics of heavy metals during the curing process. The simulated acid rain could make significant impacts on the leaching behavior of the heavy metals in the S/S materials. All cumulative fraction of leached heavy metals were less than 2.0% which showed the good stabilization effect of cement. Furthermore, the calculated diffusion coefficient for Cu was $1.28 \times 10^{-11} \text{ cm}^2/\text{s}$, indicating that its mobility was relatively

low of heavy in S/S sediments.

Appendix C

General Response to Editors and Reviewers:

Dear Editor:

I am grateful for the opportunity of the revision, which can greatly improve the manuscript.

We have carefully reviewed the editor's comments and reviewer's question, and try our best to respond to every question one by one. After response, the changes were made in the manuscript.

Checklist files of this revision:

- ◆ A response letter addressed to all the questions.
- ◆ A clean version of manuscript, with changes to the question addressed
- ◆ A version with track changes.
- ◆ The Support information with the original data for Fig. 3, Fig. 5 and the XRD and SEM results.

1.Editors:

Q1. Table 4 reports no stabilisation rate (contrary to what stated in its caption). I tried to calculate the stabilisation rates using the formula provided in the text, but the values came out different to those reported in the text. I also found the definition of the sigma value in the text confusing (e.g., C_x should be a concentration and not a total amount and I guess it is “the concentration of metal X in the leachate of the solidified body” and not “the leaching amount of heavy metal X in solidified body($\text{mg}\cdot\text{kg}^{-1}$)”).

Response:

The errors about stabilization rate were corrected in the original article and Table 4.

Changes: Section 3.1 one paragraph before Table 4, and Table 4. (Page 9)

The stabilization rates of Cd and Cu under standard curing condition were 94.6% and 79.8%, respectively. The order of heavy metals' stabilization rates was still $\text{Pb} > \text{Cd} > \text{Cu}$ of solidified body after rainfall experiments. Pb was not detected in the leaching test after rainfall either. The stabilization rates of Cd and Cu after rainfall test were 92.6% and 78.5%, respectively. The stabilization rates of Cd and Cu were decreased by 2% and 1.3%, respectively.

Table 4 Stabilization rates of leached Cd^{2+} , Pb^{2+} and Cu^{2+} with different curing conditions. (The percentage indicates the heavy metal stabilization rate after 28 days.)

Heavy metal content	Cd	Cu	Pb
The tested sediments (mg/kg)	1.11	60.4	156
Standard curing condition (mg/kg)	0.0604	12.2	-
Stabilization rates (%)	94.6	79.8	-
After rainfall test (mg/kg)	0.0825	13.0	-
Stabilization rates (%)	92.6	78.5	-

- means the concentration is below detection limit.

As for the definition of σ , additions have been made to the original article:

where σ is the stabilization rate of heavy metal (%), C_x is the leaching amount of heavy metal X in per unit mass solidified sample($\text{mg}\cdot\text{kg}^{-1}$), C is the total amount of heavy metal X in per unit mass raw sediment ($\text{mg}\cdot\text{kg}^{-1}$).

Q2. In Table 6 and 7 the D_e and i values are reported in cm^2/sec whilst in the text are in m^2/sec . That is quite confusing and make it difficult to follow the reasoning of the discussion, please homogenise.

Response:

Related errors in the text have been fixed in the original article. The unit of cm^2/s was used for consistency

Changes: Section 3.2.5 Paragraph 3. (Page 18-19)

Table 6 Slope values of each leaching period for Cu with leachant pH=2.0

Number	Leaching period (d)	Slope value of $\log(B_t)$ versus $\log(t)$	D_e (cm^2/s)	LX^a
0.5d	-	-
1d	0.68**	-
2d	0.54	1.17×10^{-11}
3d	0.56	1.32×10^{-11}	10.89
4d	0.55	1.35×10^{-11}
5d	0.78**	-
19d	0.10*	-

* surface wash-off controlled leaching mechanism

** dissolution controlled leaching mechanism

The average value of D_e for Cu is $1.28 \times 10^{-11} \text{ cm}^2/\text{s}$. According to Malviya and Chaudhary [39], if D_e is less than $3 \times 10^{-9} \text{ cm}^2/\text{s}$, the mobility of the contaminants is relatively low. While the contaminant is prone to high mobility if D_e is higher than $1 \times 10^{-7} \text{ cm}^2/\text{s}$. The mobility of contaminants is middle when D_e is between $3 \times 10^{-9} \text{ cm}^2/\text{s}$ and $1 \times 10^{-7} \text{ cm}^2/\text{s}$. From Table 6, the mean D_e for Cu was much less than $3 \times 10^{-9} \text{ cm}^2/\text{s}$, the mobility of Cu was low.

Table 7 Diffusion coefficient (D_i) by ASTM C1308-08

	pH	Slope value of CFL versus \sqrt{t}	R^2	D_i (cm^2/s)
Cu	2.0	3.88×10^{-6}	0.9572	8.18×10^{-12}
	4.0	1.06×10^{-6}	0.9941	6.11×10^{-13}
	7.0	1.05×10^{-7}	0.9128	5.99×10^{-15}
Pb	2.0	3.80×10^{-6}	0.9301	7.85×10^{-12}
	4.0	2.33×10^{-7}	0.8922	2.95×10^{-14}
	7.0	-	-	-
Cd	2.0	1.17×10^{-5}	0.9386	7.44×10^{-11}
	4.0	-	-	-
	7.0	-	-	-

All D_i were less than $3 \times 10^{-9} \text{ cm}^2/\text{s}$, which indicated the mobility of all heavy metals was relatively low.

Q3. LX is reported in Table 6 and not in Table 5.

Response:

Related errors have been fixed in the original article.

Changes: Section 3.2.6 Paragraph 3. (Page 19)

Moreover, from Table 6, LX for Cu under strongly acidic condition was higher than 9.

2. Reviewer: 1

Q1. The revised manuscript is clearly improved, but some surprising lacks are encountered (more than typographic checking). For example, in abstract and conclusion paragraph, only the Cu behavior is presented, and not on Pb and Cd.

Response:

The reviewer's suggestion was well taken and some changes were made.

The D_e of Pb and Cd can't be calculated by Eq. (2) because their leaching mechanisms are dissolution controlled. However, the diffusion coefficient D_i of Cd, Cu and Pb can be calculated by ASTM C1308-08. Relevant data have been supplemented in the abstract and conclusion.

Changes:

In the abstract (Page 1)

The calculated diffusion coefficient (D_i) for Cd, Cu and Pb were 7.44×10^{-11} , 8.18×10^{-12} and $7.85 \times 10^{-12} \text{ cm}^2/\text{s}$ respectively, indicating that its mobility was relatively low of heavy metal in S/S sediments.

Section 4 conclusions (Page 20)

The controlling leaching mechanism of Cu under strongly acidic condition was "diffusion" according to USEPA Method 1315. The average diffusion coefficient D_e of Cu was $1.28 \times 10^{-11} \text{ cm}^2/\text{s}$, and the diffusion coefficient D_i calculated according to ASTM C1308-08 was $8.18 \times 10^{-12} \text{ cm}^2/\text{s}$, which was close to the result of USEPA Method 1315. The D_i of Cd and Pb were 7.44×10^{-11} and $7.85 \times 10^{-12} \text{ cm}^2/\text{s}$, calculated according to ASTM C1308-08 respectively. Both results demonstrated that the mobility of Cd, Cu and Pb was relatively low under strongly acidic condition.

Q2. Other example, the used assumptions shall be carefully documented: for example, the use of nitrate solutions to contaminate the sediments can delay the setting and the hardening of the cement then of the treated sediments, and influence the stabilization of the heavy metals.

Response:

The reason for selection of nitrate $M(NO_3)_2$ is that it is chemically inert as compared with chloride, sulfate and acetate ions, to react with the metal ion in cement hydration. It is in order to simulate heavy metal contaminated sediments, the influence is not studied in the article.

Changes: Section 2.1 below table1. (Page 4)

The reason for choosing nitrate is that it is chemically inert as compared with chloride, sulfate and acetate ions to react with the metal ion in cement hydration. The use of nitrate may delay the setting and hardening of cement of sediments, but the concentration is relatively low, and the effect is relatively weak.

Q3. Last example, the responses to the reviewer should be included in the manuscript. If the mechanical performance of the treated sediments decrease of 14%, what is the damaging effect on site ?

Response:

The data about UCS of standard curing condition and rainfall conditions with different days are as follows, the average rate was $(34.29+24.24+7.89-9.09)/4=14.33\%$

days	UCS of standard curing condition(kPa)	UCS of rainfall conditions(kPa)	rate(%)
3	220	240	+9.09
7	350	230	34.29
14	660	500	24.24
28	760	700	7.89

We choose 300 kPa as benchmark in the study. It complies with the recommended values under the Resource Conservation and Recovery Act (RCRA). We can see the UCS exceeds the standard after 14 and 28 days of rainfall test. But different erosion environment will affect the compressive strength. If 300kPa (7d compressive strength) is used as the standard of safe landfill, there may be a risk that the compressive strength will not reach the standard after the landfill is eroded by acid rain.

Changes: 1st paragraph after Sec 3.1. (Page 7)

The maximum deterioration rate was 34.29% of the compressive strength after 7 days of curing. The minimum rate was 7.89% after 28 days.

End of 2 paragraphs after Fig. 2. (Page 8-9)

The UCS of Standard condition of 7 days, was above 300 kPa, which is qualified. if the UCS were curing at 7 days after rainfall tests, the UCS is below 300 kPa, which is unsatisfactory. Later on, at 14

day of curing, the UCS of both standard and rainfall tests were higher than benchmark of 300 kPa. Figure 2, indicated the curing time should be longer (14d > 7 d) after rain fall tests to makes the UCS qualified. The UCS of standard and rainfall conditions were preferred at 14 day of curing or longer.

3. Reviewer: 2

Q1. The authors made some revisions to the manuscript. However, some critical issues still remain unaddressed. The novelty and academic merits of this study are not properly justified. Specific comments are provided as follows:

Response:

The novelty and academic merits of this study:

1. The 22nd world dredging conference in 2019 promoted the Shanghai consensus on the sustainable development of dredging, and proposed that dredging and its utilization of dredging mud should follow the principles of "co construction with nature" and "optimization of ecological value". For example, the utilization of dredging soil resources can make the dredging project more economical as a whole and reduce the use of new materials in production and transportation. With the development of technology, the concept of sediment as a resource has been paid more and more attention.

2. By exploring the influence of rainfall on the solidification/stabilization process and the long-term safety of solidified body, the theoretical basis and technical reference are provided for the actual ecological restoration of river sediment.

Changes: Introduction (Page 2)

It has become an important topic in the field of environmental protection to solve the secondary pollution of the sediment and make use of the dredged sediment as a resource. The 22nd world dredging conference in 2019 promoted the Shanghai consensus on the sustainable development of dredging, and proposed that dredging and its utilization of dredging mud should follow the principles of "co construction with nature" and "optimization of ecological value". For example, the utilization of dredging soil resources can make the dredging project more economical as a whole and reduce the use of new materials in production and transportation. With the development of technology, the concept of sediment as a resource has been paid more and more attention.

Q2. The authors just simply neglected the suggested references. The recent papers on solidification and

stabilization of contaminated sediment have to be critically reviewed and compared in this study, otherwise this study is clearly not built on the latest findings in the existing literature and cannot be accepted for publication. For example:

- The roles of biochar as green admixture for sediment-based construction products. *Cem. Concr. Compos.* 104 (2019) 103348
- Novel synergy of Si-rich minerals and reactive MgO for stabilisation/solidification of contaminated sediment. *J. Hazard. Mater.* 365 (2019) 695-706
- Recycling dredged sediment into fill materials, partition blocks, and paving blocks: Technical and economic assessment. *J. Clean. Prod.* 199 (2018) 69–76
- Green remediation of contaminated sediment by stabilization/solidification with industrial by-products and CO₂ utilization. *Sci. Total Environ.* 631 (2018) 1321-1327
- Recycling contaminated sediment into eco-friendly paving blocks by a combination of binary cement and carbon dioxide curing. *J. Clean. Prod.* 164 (2017) 1279-1288

Response:

The reviewer's suggestion was well taken, and we read these reference and put them into Introduction, Experiments, Result and Discussion, in related areas with [8, 9, 20, 26, 27].

Changes:

Introduction (Page 2)

Carbonation has a significant improvement on cement-based stabilization/solidification in terms of the cement hydration and mechanical strength [8-9].

Section 2.1 materials (Page 4)

Particle size distribution of sediments influenced the pore structure formation and strength development [20].

Section 3.1, 1 paragraph below Fig.2. (Page 8)

Comparing with the results of other publications, the trend is consistent [8,9,20,26-27], that UCS increase with increased curing time (1, 7, 28 d in most cases), which is 28d > 14d > 7d > 3d. The longer curing time the better under difference circumstances. However, those studies has different values as benchmark, such as 1.0 MPa [26], 30 MPa [8-9,20], 45 MPa[9,20]. Among these values, the 30 MPa was mostly used [8-9,20]. However, our study used different products, the 300 kPa was adequate for sediments. Furthermore, some studies adopted CO₂ curing replacing air curing [8-9], indicating better

UCS values with CO₂ curing. This result is useful except the cost was bigger. The UCS of CO₂ curing of this study await further study.

Q3. Fig. 2 What is the requirement of compressive strength for these products?

Response:

Unconfined compressive strength is an important index to evaluate the engineering properties of solidified body. According to the different ways of recycling dredged sediment, the relevant standards and many scholars at home and abroad put forward different unconfined compressive strength requirements. The unconfined compressive strength of the cured body after 7 days curing is generally used as the control index in foreign research, Koenig A et al. [1] suggest that the unconfined compressive strength of solid waste for landfill treatment should be greater than 50 kPa, and the Resource Conservation and Recovery Act (RCRA) of the United States suggest it should be greater than 300 kPa [2]. **We choose 300 kPa as benchmark in the study.**

Reference

- [1] Koenig A, Kay J, Wan I. Physical properties of dewatered wastewater sludge for landfilling. *Water Science & Technology*, 1996, 34(3-4):533-540.
- [2] Shi C, A. Fernández-Jiménez. Stabilization/solidification of hazardous and radioactive wastes with alkali-activated cements. *Journal of Hazardous Materials*, 2006, 137(3):1656-1663.

Changes: (Page 8, Fig.2)

Fig.2 UCS comparison of standard curing condition and rainfall conditions, 300 kPa was selected as benchmark according to Resource Conservation and Recovery Act (RCRA) of the United States

Q4. Line 46-48: The authors said gypsum and ettringite were generated. Please provide direct evidence.

Response:

These products were expected to generate based on the chemical reaction, which are listed below:

Unfortunately, our experimental results cannot discover these products so far, therefore, the original sentence was removed.

Changes: Section 3.2.4 (Page 17)

Based on the reactions below, the gypsum and ettringite might be generated. However, whether these products were generated in the sediments, await detailed experiments.

Appendix D

General Response to Editors and Reviewers:

Dear Editor:

I am grateful for the opportunity of the revision, which can greatly improve the manuscript. We have carefully reviewed reviewer's question, and try our best to respond to every question. The changes were made in the right place of the manuscript.

Checklist files of this revision:

Reviewer 1

Q1. The carbonation is a pathology of the cementitious materials. It cannot be identify as an S/S process improvement.

Response:

The reviewer's suggestion was taken, and some changes were made.

Changes:

In the introduction (Page 2)

The production of carbonation has its positive effect on the mechanical strength of the solidified sediments when using the cement based S/S Carbonation has a significant effect improvement on cement-based stabilization/solidification in terms of the cement hydration and mechanical strength [8-9]

Q2. It is necessary to precise that the samples contain only sand and powder, but little clay minerals and secondary minerals. The experimental sediment consists of 3.7% clay particles, 70.9% silt particles, 25.4% sand particles in Table 1 for example. So, the sediment is defined as a silt.

Response:

The reviewer's suggestion was well taken and some changes were made.

Changes: (section 2.1 , page 4)

Sediment textures can influenced the pore structure formation and strength development [20]. In this study, the particle size distribution. The particle size distribution was measured by the intelligent laser particle size analyzer (BT-9300z, Baite). According to particle size, the experimental The tested

sediment consists of 3.7% clay particles, 70.9% silt particles, 25.4% sand particles and is therefore classified as silty mud. So, the sediment is defined as a silt.

Q3. The addition concentration is based on China's specification "GB 15618-2018. Soil environment quality risk control standard for soil contamination of agriculture land".

Response:

We have added the reference standard for sample configuration in the manuscript.

Changes: (section 2.1 , page 4)

To prepare contaminated sediment specimens, predetermined volume of $\text{Pb}(\text{NO}_3)_2$, $\text{Cu}(\text{NO}_3)_2 \cdot 3\text{H}_2\text{O}$ and $\text{Cd}(\text{NO}_3)_2 \cdot 4\text{H}_2\text{O}$ solution were added to the air-dried sediments until its water content reached 45%. The concentrations of the heavy metals added were based on the soil environmental quality risk control standard for soil contamination of agricultural land (China, GB 15618—2018).

Q4. The reference to justify nitrate solution addition is not relevant.

Response:

The contents are not related to the references, and the reference was replaced with the original one.

Changes:

[21] Xia W Y, Feng Y S , Jin F , et al. Stabilization and solidification of a heavy metal contaminated site soil using a hydroxyapatite based binder. *Construction and Building Materials*, 2017, 156:199-207.

Q5. Only Cu, Cd and Pb are studied: why not Cr, Ni and Zn, which are also analyzed ? The answer is to insert in the manuscript.

Response:

Cd, Cu and Pb are very representative in heavy metal pollution, so they are selected. Cr, Ni and Zn are also representative and some content was provided in Table 2. The answer has been inserted in the manuscript.

Changes:(Section 2.1, page 4)

We chose Cd, Cu and Pb as the research objects since they are representative in heavy metal pollution. Other metals, Cr, Ni and Zn are also representative with the content provided in Table 2.

Q6. In the manuscript, to add how much samples are tested for each configuration of test (UCS and leaching tests), what is the repeatability of the tests : we prepared three groups of parallel samples and took the average values. The error bars were shown in the chart of Figure 2 and 5.

Response:

The reviewer’s suggestion was well taken, and we provided specific experimental conditions and repeated times in table 4.

Changes: (Section 2.2.2 , page 8)

Table 4. Specific experimental conditions and test times for different tests

Test type	Quality ratio of cement to the sediment	Curing time (days)	Number of identical samples	Number of replicates
Field UCS		3, 7, 14, 28	3	1
rainfall test	TCLP 1:4	3, 7, 14, 28	3	3
Semi-dynamic leaching test		28	1	3

Q7. In abstract, introduction and conclusion, to indicate the objective of reuse of the sediments : landfill.

According to the Chinese design code “regulations in the design code for highway subgrade (JTG d30-2015)”, the UCS index of the lightweight soil of the subgrade should be greater than 600 kPa for expressways and first-class highways, and more than 500 kPa for secondary and sub-secondary highways. The UCS of the solidified sediment meets the specification requirements, so the heavy metal contaminated sediment can be used as roadbed filler after solidified and stabilized.

Koenig A [1] suggest that the UCS of waste should be greater than 50 kPa for landfill treatment, while the resource conservation and recovery act (RCRA) of the United States suggests that it should be greater than 300 kPa [2]. In this study, the recommended value of RCRA is 300 kPa, so the curing energy can meet the requirement of building backfill strength.

[1] Koenig A, Kay J N, Wan I M. Physical properties of dewater sludge for landfilling [J]. Water Science and Technology, 1996, 34:533-540.

[2] Shi C, Fernández-Jiménez A. Stabilization/solidification of hazardous and radioactive wastes with alkali-activated cements [J]. Journal of hazardous materials, 2006, 137(3): 1656-1663.

Response:

The reviewer’s suggestion is well taken. The abstract, introduction and conclusion indicate that the application of sediment based materials is a landfill, we added the application standard of roadbed backfilling.

Changes:

The Resource Conservation and Recovery Act (RCRA) of the United States suggest the unconfined compressive strength of solid waste for landfill treatment should be greater than 300 kPa[28]

[28] Shi C, A. Fernández-Jiménez. Stabilization/solidification of hazardous and radioactive wastes with alkali-activated cements. Journal of Hazardous Materials, 2006, 137(3):1656-1663.

Q8. To precise that, taking the strength qu_0 at the ratio of height to diameter of 2 as the reference value, through regression analysis, the ratio of qu to qu_0 at any ratio of height to diameter of H/D is obtained as follows:

$$qu / qu_0 = 0.983(H/2D)^{-0.455}$$

Response:

The UCS data have been converted to the standard value of height to diameter ratio 2:1, to compare with the standard.

The Fig.2 was updated in the manuscript.

Changes: (section 3.1)

Fig.2 UCS comparison of standard curing condition and rainfall conditions ,300 kPa was selected as benchmark according to Resource Conservation and Recovery Act (RCRA) of the United States

Q9. To indicate "Malviya and Chaudhary pointed out that when the observation diffusion coefficient $Diobs < 3$

$\times 10^{-13}$ m²/s, the pollutant migration is low. When 3×10^{-13} m²/s $< D_i < 10^{-11}$ m²/s, the pollutant migration is moderate. When $D_i > 10^{-11}$ m²/s, pollutant migration is high.

In order to interpret the results, the negative log of observation diffusion coefficient ($-\lg D_i$) can be taken to define the Leachability index of heavy metals after correction. Environment Canada has listed this parameter as a control index for the utilization and disposal of solidified and stabilized contaminated soils. The formula for LX is shown below. When LX is greater than 9, it indicates that the curing and stabilization treatment of contaminated soil is effective and can be used for controlling the curing stop, such as as roadbed filler. When LX is greater than 8, the solidified soil can be disposed of by landfill. When LX is less than 8, the solidified soil is not suitable for landfill disposal."

Response:

The LX has been inserted in the manuscript in a column in Table 7.

Changes:(section 3.2.6,page 19)

Table 7 Diffusion coefficient (D_i) by ASTM C1308-08

	pH	Slope value of CFL versus \sqrt{t}	R ²	D_i (cm^2/s)	LX= $-\log(D_i)$
Cu	2.0	3.88×10^{-6}	0.9572	8.18×10^{-12}	11.09
	4.0	1.06×10^{-6}	0.9941	6.11×10^{-13}	12.21
	7.0	1.05×10^{-7}	0.9128	5.99×10^{-15}	14.22
Pb	2.0	3.80×10^{-6}	0.9301	7.85×10^{-12}	11.11
	4.0	2.33×10^{-7}	0.8922	2.95×10^{-14}	13.53
	7.0	-	-	-	-
Cd	2.0	1.17×10^{-5}	0.9386	7.44×10^{-11}	10.13
	4.0	-	-	-	-
	7.0	-	-	-	-

Q10. Are specific form of erosion observed on the samples/specimens surface? does the leaching reached the sample/specimen core (sufficient porosity or permeability)?

All of the previous studies suggest that diffusion is the controlling mechanism of most heavy metals that are leaching from S/S treated sediments including this research [1] ,which is not involved in the specimen core.

[1] Yan-Jun Du, Ming-Li Wei, Krishna R. Reddy, Zhao-Peng Liu, Fei Jin. Effect of acid rain pH on leaching behavior of cement stabilized lead-contaminated soil. Journal of Hazardous Materials, 2014, 271.

Response:

Yes, from Fig. 4 the erosion was observed on the sample/specimens surface at pH=2.0, but it was not observed at

pH=4.0 and 7.0. Because our leaching time was 34.5 days, the leaching reached the sample center. After the test, we crushed the sample of (pH 2.0) and found that all the samples had been soaked.

Appendix E

General Response to Editors and Reviewers:

Dear Editor:

1. I am grateful for the opportunity of this final revision. We revised the discussion of LX parameters were completed.
2. The Table and Figures were provided in the application package.
3. The high-resolution figures in PDF format were uploaded separately.
4. The updated raw data that support the Result Figures 2, 3, 5 were also uploaded in ESM.
5. The overall manuscript was proofread twice with an English native speaker and revised again properly.

Main revisions:

Associate Editor Comments to Author (Dr Riccardo Avanzinelli):

I strongly recommend having the manuscript checked and proof-read by an English mother tongue or through professional English language editing.

Response: **The manuscript was thoroughly proofread twice by a native tongue writer, and it was revised again to improve the quality.**

Also, I am not convinced by the authors' response to one of the reviewers comments (Q9). The reviewer suggested a sentence to explain the use of the leachability index [$LX = -\log(D_i)$], and the interpretation of its values. The authors added the LX parameter in the Table 7 (now Table 8), but did not modify the sentence at all. I strongly suggest to modify the sentence according to the reviewer suggestion, or at least to explain a bit better the use of the LX parameters, whose values are now provided in Tables 7 and 8, without too much discussion.

Response:

The reviewer's suggestion was taken, the use of LX and its interpretations need to be discussed. It was provided in both "3.2.1 Theory" and "3.2.6 result and discussion"

Changes:**Theory: Line 2 below equation 3:**

The effectiveness of immobilization of the heavy metals in the S/S monolith was evaluated by the leachability index (LX), which is the negative log of observation diffusion coefficient ($-\lg D_{e \text{ obs}}$). According to Environment Canada [36], LX is a performance criterion for the utilization and disposal of S/S treated waste, which can intuitively reflect the migration of heavy metals in S/S treated waste.

Line 2 below Table 8:

From Table 8, all the LX values were bigger than 9, which indicated the treatment process of heavy metals were effective and S/S wastes can be safely used in area of road base, quarry rehabilitation, and lagoon closure.